

# Evaluating Precipitation Distributions at Regional Scales: A Benchmarking Framework and Application to CMIP 5 and 6 Models

Min-Seop Ahn[1,*], Paul A. Ullrich[2,1], Peter J. Gleckler[1], Jiwoo Lee[1], Ana C. Ordonez[1], and Angeline G. Pendergrass[3,4]

[1]*PCMDI, Lawrence Livermore National Laboratory, Livermore, CA, USA*
[2]*Department of Land, Air and Water Resources, University of California, Davis, CA, USA*
[3]*Earth and Atmospheric Science, Cornell University, Ithaca, NY, USA*
[4]*National Center for Atmospheric Research, Boulder, CO, USA*

\* Corresponding author: Min-Seop Ahn (ahn6@llnl.gov)



**Abstract**
A framework for quantifying precipitation distributions at regional scales is presented and
applied to CMIP 5 and 6 models. We employ the IPCC AR6 climate reference regions
over land and propose refinements to the oceanic regions based on the homogeneity of
precipitation distribution characteristics. The homogeneous regions are identified as
heavy, moderate, and light precipitating areas by K-means clustering of IMERG
precipitation frequency and amount distributions. With the global domain partitioned into
62 regions, including 46 land and 16 ocean regions, we apply 10 established precipitation
distribution metrics. The collection includes metrics focused on the maximum peak, lower
10th percentile, and upper 90th percentile in precipitation amount and frequency
distributions, the similarity between observed and modeled frequency distributions, an
unevenness measure based on cumulative amount, average total intensity on all days
with precipitation, and number of precipitating days each year. We apply our framework
to 25 CMIP5 and 41 CMIP6 models, and 6 observation-based products of daily
precipitation. Our results indicate that many CMIP 5 and 6 models substantially
overestimate the observed light precipitation amount and frequency as well as the number
of precipitating days, especially over mid-latitude regions outside of some land regions in
the Americas and Eurasia. Improvement from CMIP 5 to 6 is shown in some regions,
especially in mid-latitude regions, but it is not evident globally, and over the tropics most
metrics point toward over degradation.



## 1.  Introduction

Precipitation is a fundamental characteristic of the Earth's hydrological cycle and one that
can have large impacts on human activity. The impact of precipitation depends on its
intensity and frequency characteristics (e.g., Trenberth et al. 2003; Sun et al. 2006;
Trenberth and Zhang 2018). Even with the same amount of precipitation, more intense
and less frequent rainfall is more likely to lead to extreme precipitation events such as
floods and drought compared to less intense and more frequent rainfall. While mean
precipitation has improved in Earth system models, the precipitation distributions continue
to have biases (e.g., Dai 2006; Fiedler et al. 2020), which limits the utility of these
simulations, especially at the level of accuracy that is increasingly demanded in order to
anticipate and adapt to changes in precipitation due to global warming.
Multi-model intercomparison with a well-established diagnosis framework facilitates
identifying common model biases and sometimes yields insights into how to improve
models. The Coupled Model Intercomparison Project (CMIP; Meehl et al. 2000, 2005,
2007; Taylor et al. 2012; Eyring et al. 2016) is a well-established experimental protocol to
intercompare state-of-the-art Earth system models, and the number of models and
realizations participating in CMIP has been growing through several phases from 1
(Meehl et al. 2000) to 6 (Eyring et al. 2016). Given the increasing number of models,
developed at higher resolution and with increased complexity, modelers and analysts
could benefit from capabilities that help synthesize the consistency between observed
and simulated precipitation. Pendergrass et al. (2020) envisioned a framework for both
baseline and exploratory precipitation benchmarks, and Leung et al. (2022) described



efforts to advance exploratory objective evaluation for simulated precipitation focusing on
process-oriented and phenomena-based metrics at a variety of spatiotemporal scales.
The baseline precipitation benchmark metrics target established measures of the mean
state, the seasonal and diurnal cycles, variability across timescales, intensity/frequency
distributions, extremes, and drought. The current study provides a framework focused on
precipitation distributions.
Diagnosing precipitation distributions and formulating metrics that extract critical
information from precipitation distributions have been addressed in many previous studies.
Pendergrass and Deser (2017) proposed several precipitation distribution metrics based
on frequency and amount distribution curves. The precipitation frequency distribution
quantifies how often rain occurs at different rain rates, whereas the precipitation amount
distribution quantifies how much rain falls at different rain rates. Based on the distribution
curves, Pendergrass and Deser (2017) extracted rain frequency peak and amount peak
where the maximum non-zero rain frequency and amount occur, respectively.
Pendergrass and Knutti (2018) introduced a metric that measures the unevenness of daily
precipitation based on the cumulative amount curve. Their unevenness metric is defined
as the number of wettest days that constitute half of the annual precipitation. In the
median of station observations equatorward of 50° latitude, half of the annual precipitation
falls in only about the heaviest 12 days, and generally models underestimate the
observed unevenness (Pendergrass and Knutti 2018). In addition, several metrics have
been developed to distill important precipitation characteristics, such as the fraction of
precipitating days and simple daily intensity index (SDII, Zhang et al. 2011). In this study





we implement all these well-established metrics and several other complementary metrics
into our framework.

Many studies have analyzed the precipitation distributions over large domains (e.g., Dai
2006; Pendergrass and Hartmann 2014; Ma et al. 2022). Often, these domains comprise
both heavily precipitating and dry regions. Given the emphasis on regional scale analysis
continues to grow as models' horizontal resolution increases, interpretation of domain-
averaged distributions could be simplified by defining regions that are not overly complex
or heterogeneous in terms of their precipitation distribution characteristics. Iturbide et al.
(2020) has identified climate reference regions that have been adopted in the sixth
assessment report (AR6) of the Intergovernmental Panel on Climate Change (IPCC). Our
framework is based on these IPCC AR6 reference regions for objective examination of
precipitation distributions over land. Over the ocean we have revised some of the regions
of Iturbide et al. (2020) to better isolate homogeneous precipitation distribution
characteristics.

In this study, we propose a framework for regional scale quantification of simulated
precipitation distributions and evaluate CMIP 5 and 6 models with multiple observations.
The remainder of the paper is organized as follows: Sections 2 and 3 describe the data
and analysis methods. Section 4 presents results including the application and
modification of IPCC AR6 climate reference regions, evaluation of CMIP models, and
their improvement across generations. Sections 5 and 6 discuss and summarize the main
accomplishments and findings from this study.







## 2. Data

### 2.1. Observational data

For reference data, we use six global daily precipitation products first made available as part of the Frequent Rainfall Observations on GridS (FROGS) database (Roca et al., 2019) and then further aligned with CMIP output via the data specifications of the Observations for Model Intercomparison Project (Obs4MIPs, Waliser et al. 2020). These include five satellite-based products and a recent atmospheric reanalysis product. The satellite-based precipitation products include the Integrated Multi-satellitE Retrievals for GPM version 6 final run product (Huffman et al. 2020; hereafter IMERG), the Tropical Rainfall Measuring Mission Multi-satellite Precipitation Analysis 3B42 version 7 product (Huffman et al. 2007; hereafter TRMM), the bias-corrected Climate Prediction Center Morphing technique product (Xie et al. 2017; hereafter CMORPH), the Global Precipitation Climatology Project 1DD version 1.3 (Huffman et al. 2001; hereafter GPCP), and Precipitation Estimation from Remotely Sensed Information using Artificial Neural Networks (Ashouri et al. 2015; hereafter PERSIANN). The reanalysis product included for context is the ECMWF's fifth generation of atmospheric reanalysis (Hersbach et al. 2020; hereafter ERA5). Table 1 summarizes the observational datasets with the data source, coverage of domain and period, resolution of horizontal space and time frequency, and references. We use the data periods available via FROGS and Obs4MIPs as follows: 2001-2020 for IMERG, 1998-2018 for TRMM, 1998-2012 for CMORPH, 1997-2020 for GPCP, 1984-2018 for PERSIANN, and 1979-2018 for ERA5.






2.2.    CMIP model simulations

We analyze daily precipitation from all realizations of AMIP simulations available from
CMIP5 (Taylor et al. 2012) and CMIP6 (Eyring et al. 2016). We have chosen to
concentrate our analysis on AMIP simulations rather than the coupled Historical
simulations because the simulated precipitation in the latter is strongly influenced by
biases in the modeled sea surface temperature, complicating any interpretation regarding
the underlying causes of the precipitation errors. Table 2 lists the participating models,
the number of realizations, and the horizontal resolution in each modeling institute. We
evaluate the most recent 20 years (1985-2004) that both CMIP 5 and 6 models have in
common for a fair comparison with satellite-based observations.


**3.    Methods**
In our framework we apply 10 metrics that characterize different and complementary
aspects of the intensity distribution of precipitation at regional scales. Table 3 summarizes
the metrics including their definition, purpose, and references. The computation of the
metrics has been implemented and applied in an open source metrics package, the
Program for Climate Model Diagnosis & Intercomparison (PCMDI) metrics package (PMP;
Gleckler et al. 2008, 2016).

3.1.    Frequency and amount distributions





Following Pendergrass and Hartmann (2014) and Pendergrass and Deser (2017), we use
logarithmically-spaced bins of daily precipitation to calculate both the precipitation
frequency and amount distributions. Each bin is 7% wider than the previous one, and the
smallest non-zero bin is centered at 0.03 mm/day. The frequency distribution is the
number of days in each bin normalized by the total number of days, and the amount
distribution is the sum of accumulated precipitation in each bin normalized by the total
number of days. Based on these distributions (Fig. 1a), we identify the rain rate where the
maximum peak of the distribution appears (Amount/Frequency Peak, Pendergrass and
Deser 2017; also called mode, Kooperman et al., 2016) and formulate several
complementary metrics that measure the fraction of the distribution lower 10 percentile
(P10) and upper 90 percentile (P90). The precipitation bins less than 0.1 mm/day are
considered dry for the purpose of these calculations. The threshold rain rates for 10th and
90th percentiles are defined from the amount distribution as determined from
observations. Here we use IMERG as the default reference observational dataset. The
final frequency related metric we employ is the Perkins score, which measures the
similarity between observed and modeled frequency distributions (Perkins et al. 2007).
With the sum of a frequency distribution across all bins being unity, the Perkins score is
defined as the sum of minimum values between observed and modeled frequency across
all bins: $Perkins\ Score = \sum_1^n minimum(Z_o, Z_m)$ where $n$ is the number of bins, $Z_o$ and
$Z_m$ are the frequency in a given bin for observation and model, respectively. The Perkins
score is a unitless scalar varying from 0 (low similarity) to 1 (high similarity).

3.2.    Cumulative fraction of annual precipitation amount



Following Pendergrass and Knutti (2018), we calculate the cumulative sum of daily
precipitation each year sorted in descending order (i.e., wettest to driest) and normalized
by the total precipitation for that year. From the distribution for each individual year (see
Fig. 1b), we obtain the metrics gauging the number of the wettest days for half of annual
precipitation (Unevenness, Pendergrass and Knutti 2018) and the fraction of the number
of precipitating (>=1mm/day) days (FracPRdays). To facilitate comparison against longer-
established analyses (e.g., ETCCDI, Zhang et al., 2011), we include the daily
precipitation intensity, calculated by dividing the annual total precipitation by the number
of precipitating days (SDII, Zhang et al. 2011). To obtain values of these metrics over
multiple years, we take the median across years following Pendergrass and Knutti (2018;
for unevenness).

3.3.    Reference regions

We use the spatial homogeneity of precipitation characteristics as a basis for defining
regions, as in previous studies (e.g., Swenson and Grotjahn 2019). In addition to
providing more physically-based results, this also simplifies interpretation with robust
diagnostics when we average a distribution characteristic across the region. We use K-
means clustering (MacQueen 1967) with the concatenated frequency and amount
distributions of IMERG over the global domain to identify homogeneous regions for
precipitation distributions. K-means clustering is an unsupervised machine learning
algorithm that separates characteristics of a given dataset into a specified number of
groups, which has been widely used because it is faster and simpler than other methods.
Figure 2 shows the spatial pattern of IMERG precipitation mean state and clustering



results with 3 clusters identified by the algorithm (Fig. 2b) including heavy (blue),
moderate (green), and light (orange) precipitation regions. The spatial pattern of these
clustering regions resembles the pattern of the mean state of precipitation, providing a
sanity check indicating that the cluster-based regions are physically reasonable.

In support of the AR6, the IPCC proposed a set of climate reference regions (Iturbide et
al. 2020). These regions were defined based on geographical and political boundaries
and the climatic consistency of temperature and precipitation in current climate and
climate change projections. When defining regions, the land regions use both information
from current climate and climate change projections, while the ocean regions use only
the information from climate change projections. In other words, the climatic consistency
of precipitation in the current climate is not explicitly represented in the definition of the
oceanic regions. Figure 3a shows the IPCC AR6 climate reference regions superimposed
on our precipitation clustering regions shown in Fig. 2b. The land regions correspond
reasonably well to the clustering regions, but some ocean regions are too broad, including
both heavy and light precipitating regions (Fig. 3a). In this study, the ocean regions are
modified based on the clustering regions, while the land regions remain the same as in
the AR6 (Fig. 3b).

In the Pacific Ocean region, the original IPCC AR6 regions consist of equatorial Pacific
Ocean (EPO), northern Pacific Ocean (NPO), and southern Pacific Ocean (SPO). Each
of these regions includes areas of both heavy and light precipitation. EPO includes the
Intertropical Convergence Zone (ITCZ), the South Pacific Convergence Zone (SPCZ),



and also the dry southeast Pacific region. The NPO region includes the north Pacific storm
track and the dry northeast Pacific. The SPO region includes the southern part of SPCZ
and the dry southeast area of the Pacific. In our modified IPCC AR6 regions, the Pacific
Ocean region is divided into four heavy precipitating regions (NPO, NWPO, PITCZ, and
SWPO) and two light and moderate precipitating regions (NEPO and SEPO). The NPO,
NWPO, PITCZ, and SWPO mainly include the North Pacific storm track region, the
western Pacific warm pool region, pacific ITCZ, and SPCZ, respectively. The NEPO and
SEPO respectively include the northeast and southeast dry Pacific regions. Similarly, in
the Atlantic Ocean region, the original IPCC AR6 regions consist of the equatorial Atlantic
Ocean (EAO), northern Atlantic Ocean (NAO), and southern Atlantic Ocean (SAO), with
each including both heavy and light precipitating regions. Our modified Atlantic Ocean
region consists of two heavy precipitating regions (NAO and AITCZ) and two light and
moderate precipitating regions (NEAO and SAO). The NAO and AITCZ mainly include
the North Atlantic storm track region and Atlantic ITCZ, respectively. The NEAO and SAO
mainly include dry eastern Atlantic regions. The Indian Ocean (IO) region is not modified
as the original IPCC AR6 climate reference region separates well the heavy precipitating
equatorial IO (EIO) region from the moderate and light precipitating southern IO (SIO)
region. The Southern Ocean (SOO) is modified to mainly include the heavy precipitation
region around the Antarctic. The original IPCC AR6 climate reference regions consist of
58 regions including 12 oceanic regions and 46 land regions, while our modification
consists of 62 regions including 16 oceanic regions and the same land regions as the
original (see Table 4). Note that the Caribbean (CAR), the Mediterranean (MED), and
Southeast Asia (SEA) are not counted for the oceanic regions.






### 3.4. Evaluating model performance

We use two simple measures to compare the collection of CMIP 5 and 6 model
simulations with the five satellite-based observational products (IMERG, TRMM,
CMORPH, GPCP, and PERSIANN). One gauges how many models within the multi-
model ensemble fall within the observational range between the minimum and maximum
observed values for each metric and each region. Another is how many models
underestimate or overestimate all observations, i.e., are outside the bounds spanned by
the minimum and maximum values across the five satellite-based products. To quantify
the dominance of underestimation versus overestimation of the multi-model ensemble
with a single number, we use the following measure formulation: $(nO - nU)/nT$ where $nO$
is the number of overestimating models, $nU$ is the number of underestimating models,
and $nT$ is the total number of models. Thus, positive values represent overestimation, and
negative values represent underestimation. If models are mostly within the observational
range or widely distributed from underestimation to overestimation, the quantification
value would approach zero.

Many metrics that can be used to characterize precipitation, including those used here,
are sensitive to the spatial and temporal resolutions at which the model and observational
data are analyzed (e.g., Pendergrass and Knutti 2018, Chen and Dai 2019). As in many
previous studies the diagnosis of precipitation in CMIP 5 and 6 models (e.g., Fiedler et al.
2020; Tang et al. 2021; Ahn et al. 2022), to ensure appropriate comparisons, we conduct
all analyses at a common horizontal grid of 2x2 degrees with a conservative regridding



method. For models with multiple ensemble members, we first compute the metrics for
all available realizations and then average the results across the realizations.


**4.    Results**

4.1.    Homogeneity within reference regions

For the regional scale analysis, we employ the IPCC AR6 climate reference regions
(Iturbide et al. 2020) while we revise the region dividings over the oceans based on
clustered precipitation characteristics as described in section 3.3. To quantitatively
evaluate the homogeneity of precipitating distributions in the reference regions, we use
three homogeneity metrics: the Perkins score (Perkins et al. 2007), Kolmogorov–Smirnov
test (K-S test, Chakravart et al. 1967), and Anderson-Darling test (A-D test, Stephens
1974). The three metrics measure the similarity between the regionally-averaged and
individual grid cell frequency distributions within the region. The Perkins score measures
the overall similarity between two frequency distributions, which is one of our distribution
performance metrics described in Section 3.1. The K-S and A-D tests focus more on the
similarity in the center and the side of the frequency distribution, respectively. The three
homogeneity metrics could complement each other as their main focuses are on different
aspects of frequency distributions.

In the original IPCC AR6 reference regions, the oceanic regions show relatively low
homogeneity of precipitating characteristics compared to land regions (Fig. 4). The Pacific
and Atlantic Ocean regions show much lower homogeneity than the Indian Ocean,



274 especially in EPO and EAO regions. In the modified oceanic regions, the homogeneities

275 show an overall improvement with the three homogeneity metrics. In particular, the

276 homogeneity over the heavy precipitating regions where the homogeneity was lower (e.g.,

277 Pacific and Atlantic ITCZ and mid-latitude storm track regions) are largely improved. The

278 clustering regions shown here are obtained based on IMERG precipitation. However,

279 since different satellite-based products show substantial discrepancies in precipitation

280 distributions, it is important to assess whether the improved homogeneity in the modified

281 regions is similarly improved across other different datasets. Figure 5 shows the

282 homogeneity of precipitation distribution characteristics for different observational

283 datasets using the Perkins score. Although the region modifications we have made are

284 based on the clustering regions of IMERG precipitation, Fig. 5 suggests that the

285 improvement of the homogeneity over the modified regions is consistent across different

286 observational datasets. We further tested the homogeneity for different seasons (see Fig.

287 S1 in the supplement material). The homogeneity is overall improved in the modified

288 regions across the seasons even though we defined the reference regions based on

289 annual data.

291   4.2. Regional evaluation of model simulations against multiple observations

292 The precipitation distribution metrics used in this study are mainly calculated  from three

293 curves: amount distribution, frequency distribution, and cumulative amount fraction

294 curves. Figure 6 shows these curves for three selected regions based on the clustered

295 precipitating characteristics (NWPO, which is a heavy precipitation dominated ocean

296 region; SEPO, a light precipitation dominated ocean region; and ENA, a heavy



precipitation dominated land region). The heavy and light precipitating regions are well
distinguished by their overlaid distribution curves. The amount distribution has a
distinctive peak in the heavy precipitating region (Figs. 6a and 6g), while it is flatter in the
light precipitating region (Fig. 6d). The frequency distribution is more centered on the
heavier precipitation side in the heavy precipitating region (Figs. 6b, 6h) than in the light
precipitating region (Fig 6e). The cumulative fraction increases more steeply in the light
precipitating region (Fig. 6f) than in the heavy precipitating region (Figs. 6c and 6i),
indicating there are fewer precipitating days in the light precipitating region. NWPO and
SEPO were commonly averaged for representing the tropical ocean region in many
studies, but these different characteristics in the precipitation distributions demonstrate
the additional information available via a regional scale analysis. Although satellite-based
observations are less reliable over the light precipitating ocean regions (e.g., SEPO), the
differences between heavy and light precipitation regions are well distinguishable.

In the precipitation frequency distribution, many models show a bimodal distribution in the
heavy precipitating tropical ocean region (Fig. 6b) but not in the light precipitating
subtropical ocean region (Fig. 6e) or the heavy precipitating mid-latitude land region (Fig.
6h). The bimodal frequency distribution is a commonly found in models and is seemingly
independent of resolution (e.g., Lin et al. 2013; Kooperman et al. 2018; Chen et al. 2021;
Ma et al. 2022; Martinez-Villalobos et al. 2022). It is not generally found in satellite-based
observational datasets, but this could be because the range of sensitivity to precipitation
rates is too narrow. Ma et al. (2022) compared the frequency distributions in AMIP and
HighResMIP (High Resolution Model Intercomparison Project, Haarsma et al. 2016) from





CMIP6 and DYAMOND (DYnamics of the Atmospheric general circulation Modeled On
Non-hydrostatic Domains, Satoh et al. 2019; Stevens et al. 2019) models, where they
showed that the bimodal frequency distribution appears in many AMIP (~100km),
HighResMIP (~50km), and even DYAMOND (~4km) models. Convective
parameterizations have been speculated as a cause of the light rain frequency peak (Lin
et al. 2013; Kooperman et al. 2018; Chen et al. 2021), though some models show a
convective precipitation peak at heavier precipitation than the peak from large-scale
precipitation (Martinez-Villalobos et al. 2022). ERA5 reanalysis also shows a bimodal
frequency distribution (Fig. 6b), which is not surprising considering that the reproduced
precipitation in ERA5 heavily depends on the model, thus exhibits this common model
behavior. Because of the heavy reliance on model physics to generate its precipitation
(as opposed to fields like wind, for which observations are directly assimilated), in this
study we do not include ERA5 precipitation among the observational products used for
model evaluation.

Based on the precipitation amount, frequency, and cumulative amount fraction curves,
we calculate 10 metrics (Amount peak, Amount P10, Amount P90, Frequency peak,
Frequency P10, Frequency P90, Unevenness, FracPRdays, SDII, and Perkins score) as
described in Section 3. Figure 7 shows the metrics with the modified IPCC AR6 climate
reference regions for satellite-based observations (black), ERA5 (gray), CMIP5 (blue),
and CMIP6 (red) models. The metric values vary widely across regions, especially in
Amount peak, Frequency peak, Unevenness, FracPRdays, and SDII, demonstrating the
additional detail provided by regional-scale precipitation-distribution metrics. In terms of



the metrics based on the amount distribution (Fig. 7a-c), many models tend to simulate
an Amount peak that is too light, an Amount P10 that is too high, and an Amount P90 that
is too low compared to the observations, moreso in oceanic regions (regions 47-62) than
in land regions. Similarly for the metrics based on the frequency distribution (Fig. 7d-f),
many models show light Frequency peaks, overestimated Frequency P10, and
underestimated Frequency P90 compared to observations. The similarity between
frequency distribution curves (i.e., Perkins score) is higher in land regions than in ocean
regions. Also, many models overestimate Unevenness and FracPRdays and
underestimate SDII. These results indicate that overall, models simulate more frequent
weak precipitation and less heavy precipitation compared to the observations, consistent
with many previous studies (e.g., Dai 2006; Pendergrass and Hartmann 2014; Trenberth
et al. 2017; Chen et al. 2021; Ma et al. 2022).

As expected from previous work, observations disagree substantially in some regions
(e.g., polar and high latitude regions) and/or for some metrics (e.g., Amount P90,
Frequency P90). In some cases the observational spread is much larger than that of the
models. We examine the observational discrepancy or spread by the ratio between the
standard deviation of the five satellite-based observations (IMERG, TRMM, CMORPH,
GPCP, PERSIANN) and the standard deviation of all CMIP 5 and 6 models (Fig. 8). The
standard deviation of observations is much larger near polar regions and high latitude
regions compared to the models' standard deviation for most metrics, as expected from
the orbital configurations of the most relevant satellite constellations for precipitation
(which exclude high latitudes). The Amount P90 and Frequency P90 metrics show the



largest observational discrepancy among the metrics, with standard deviations of 1.5 to
3 times larger over some high latitude regions and about 3-8 times larger over polar
regions in observations compared to the models. On the other hand, Unevenness,
FracPRdays, and Amount P10 show the least observational discrepancy – the models'
standard deviation is about 2-8 times larger than for observations over some tropical and
subtropical regions; nonetheless, the standard deviation among observations is larger
over most of the high latitude and polar regions. Model evaluation in the regions with large
disagreement among observational products remains a challenge. Note that the standard
deviation of five observations would be sensitive as there are outlier observations for
some regions and metrics (e.g., many ocean regions in Amount P90). Moreover,
observational uncertainties are rarely well quantified or understood, so agreements
among observational datasets may not always allow us to rule out common errors among
observations (e.g., for warm light precipitation over the subtropical ocean).

To attempt to account for discrepancies among observational datasets in the model
evaluation framework, we use two different approaches to evaluate model performance
with multiple observations, as described in Section 3.4. The first approach is to assess
the number of models that are within the observational range. Figure 9 shows the CMIP6
model evaluation with each metric, and the regions where the standard deviation among
observations is larger than among models are stippled gray to avoid them from the model
performance evaluation. In Amount peak, some subtropical regions (e.g., ARP, EAS,
NEPO, CAU, and WSAF) show relatively good model performance (more than 70% of
models fall in the observational range), while some tropical and subtropical (e.g., PITCZ,



AITCZ, and SEPO) and polar (e.g., RAR, EAN, and WAN) regions show poor model
performance (less than 30% of models fall in observational range). For Amount P10,
many regions are poorly captured by the simulations, except for some subtropical land
regions (e.g., EAS, NCA, CAU, and WSAF). In Amount P90, most regions are uncertain
(i.e., the standard deviation among observations is larger than among models) making it
difficult to evaluate model performance, while some tropical regions near the Indo-Pacific
warmpool (EIO, SEA, NWPO, and NAU) exhibit very good model performance (more than
90% of models fall in observational range). In the Frequency metrics (peak, P10, and
P90), more regions are difficult to evaluate model performance than in Amount metrics,
while in some tropical and subtropical regions (e.g., PITCZ, SWPO, NWPO, SEA, SAO,
and NES) model performance is good. However, good model performance could
alternatively arise from a large observational range (see Fig. 7). Unevenness,
FracPRdays, SDII, and Perkins score have a smaller fraction of models within the
observational range in tropical regions than the Amount and Frequency metrics. In
particular, fewer than 10% of CMIP6 models fall within the observational range for
Unevenness and FracPRdays over some tropical oceanic regions (e.g., PITCZ, NEPO,
SEPO, AITCZ, NEAO, SAO, and SIO).

Examining the fraction of CMIP5 models falling within the range of observations, CMIP5
models have a spatial pattern of model performance similar to that of CMIP6 models (see
Fig. S2 in supplement), and the improvement from CMIP5 to CMIP6 seems subtle. We
quantitatively assess the improvement from CMIP5 to CMIP6 by subtracting the
percentage of CMIP5 from CMIP6 models falling within the range of observations (Fig.



10). For some metrics (e.g., Amount peak, Amount and Frequency P10, and Amount and
Frequency P90) and for some tropical and subtropical regions  (e.g., SEA, EAS, SAS,
ARP, and SAH), improvement is apparent. Compared to CMIP5, 5-25% more CMIP6
models fall in the observational range in these regions. However, for the other metrics
(e.g., Frequency peak, FracPRdays, SDII, Perkins score), CMIP6 models perform
somewhat worse. Over some tropical and subtropical oceanic regions (e.g., PITCZ,
NEPO, AITCZ, and NEAO), 5-25% more CMIP6 than CMIP5 models are out of the
observational range. This result is from all available CMIP5 and CMIP6 models, so it may
reflect the fact that some models are participated in only CMIP5 or CMIP6, but not both
(see Table 2). To isolate improvements that may have occurred between successive
generations of the same models, we also compared only the models that participated in
both CMIP5 and CMIP6 (see Fig. S3). Overall, the spatial characteristics of the
improvement/degradation in CMIP6 from CMIP5 is consistent, while more degradation is
apparent when we compare this subset of models, especially over the tropical oceanic
regions (e.g., PITCZ, AITCZ, NWPO, and SEPO).

The second approach to account for discrepancies among observations in model
performance evaluation is to count the number of models that are lower or higher than all
satellite-based observations for each metric and each region. Figure 11 shows the spatial
patterns of the model performance evaluation with each metric for CMIP6 models.
Underestimation is indicated by a negative sign, while overestimation is indicated by a
positive sign via the formulation described in Section 3.4. Amount peak is overall
underestimated in most regions, indicating the amount distributions in most CMIP6



models are shifted to lighter precipitation compared to observations. In many regions, more than 50% of the CMIP6 models underestimate Amount peak. In particular, over many tropical and southern hemisphere ocean regions (e.g., PITCZ, AITCZ, EIO, SEPO, SAO, and SOO), more than 70% of the models underestimate the Amount peak. The underestimation of Amount peak is accompanied by overestimation of Amount P10 and underestimation of Amount P90. More than 70% of CMIP6 models overestimate Amount P10 in many oceanic regions; especially in the southern and northern Pacific and Atlantic, the southern Indian Ocean, and Southern Ocean more than 90% of the models overestimate the observed Amount P10. For Amount P90, it appears that many models fall within the observational range; however, observational range in Amount P90 (green boxes in Fig. 7c) is large and driven primarily by just one observational dataset (IMERG), especially in ocean regions.

For the frequency-based metrics (i.e., peak, P10, and P90; Figs. 11d-f), CMIP6 models show similar bias characteristics to Amount metrics (Figs. 11a-c), although performance is better than for Amount metrics. Over some tropical (e.g., NWPO, PITCZ, and SWPO) and Eurasia (e.g., EEU, WSB, and ESB) regions, less than 10% of models fall outside of the observed range. Unevenness and FracPRdays are severely overestimated in models. More than 90% of models overestimate the observed Unevenness (Fig. 11g) and FracPRdays (Fig. 11h) globally, especially over oceanic regions, consistent with Pendergrass and Knutti (2018). Unevenness (i.e., number of the wettest days for the half of annual precipitation) and FracPRdays (i.e., fraction of the number of annual precipitating days above 1mm/day) are highly correlated to each other; correlations





between metrics will be discussed later. SDII is underestimated in many regions globally,
especially in some heavily-precipitating regions (e.g., PITCZ, AITCZ, EIO, SEA, NPO,
NAO, SWPO, and SOO). For the Perkins score, model simulations have poorer
performance in the tropics than in the mid-latitudes and polar regions. Performance by
these various metrics is generally consistent with the often-blamed too-frequent light
precipitation and too rare heavy precipitation in simulations.

The characteristics of CMIP5 compared to CMIP6 simulations (Fig. S4) show little
indication of improvement. Here we quantitatively evaluate the improvement in CMIP6
from CMIP5 for each metric and region. Figure 12 shows the difference between CMIP5
and CMIP6 in terms of the percentage of models that under- or over-estimate each metric.
In mid-latitudes, there appears to have been an improvement in performance, however in
the tropics, there appears to be more degradation. Over some heavily-precipitating
tropical regions (e.g., PITCZ, AITCZ, EIO, and NWPO), 10-25% more models in CMIP6
than in CMIP5 overestimate Amount P10, Unevenness, and FracPRdays and
underestimate/underperform on Amount peak, SDII, and Perkins score. This indicates
that CMIP6 models simulate more frequent light precipitation and less frequent heavy
precipitation over the heavily-precipitating tropical regions. Over some mid-latitude land
regions (e.g., EAS, ESB, RFE, and ENA), on the other hand, 5-20% more models in
CMIP6 than in CMIP5 simulate precipitation distributions close to observations (i.e., less
light precipitation and more heavy precipitation). To evaluate the improvement between
model generation, we also compare only the models that participated in both CMIP5 and
CMIP6 (Fig. S5) rather than all available CMIP5 and CMIP6 models. For the subset of




models participating in both generations, the improvement characteristics are similar for
all models, although more degradation is exhibited over some tropical oceanic regions
(e.g., PITCZ, NWPO, and SWPO). This also indicates that some models newly
participating in CMIP6, and not in the CMIP5, have higher than average performance.

4.3.    Correlation between metrics

Each precipitation distribution metric implemented in this study is chosen to target
different aspects of the distribution of precipitation. To the extent that precipitation
probability distributions are governed by a small number of key parameters (as argued by
Martinez-Villalobos and Neelin 2019), we should expect additional metrics to be highly
correlated. Figure 13 shows the global weighted average of correlation coefficients
between the precipitation distribution metrics across CMIP5 and CMIP6 models. Higher
correlation coefficients are found to be between Amount P90 and Frequency P90 (0.98)
and between Amount P10 and Frequency P10 (0.67). This is expected because the
amount and frequency distributions differ only by a factor of the precipitation rate (e.g.,
Pendergrass and Hartmann 2014). Another higher correlation coefficient is between
Unevenness and FracPRdays (0.77), indicating that the number of the heaviest
precipitating days for half of annual precipitation and the total number of annual
precipitating days are related. Amount and Frequency peak metrics are negatively
correlated to P10 metrics and positively correlated to P90 metrics, but the correlation
coefficients are not very high (lower than 0.62). This is because the peak metrics focus
on typical precipitation, rather than the light and heavy ends of the distribution that are
the focus of P10 and P90 metrics. SDII is more negatively correlated with Amount P10 (-



0.67) and positively correlated with Amount peak (0.61) and less so with Amount P90
(0.48), implying that SDII is mainly influenced by weak precipitation amounts rather than
heavy precipitation amounts. The Perkins score shows relatively high negative correlation
with Unevenness (-0.62), FracPRdays (-0.59), and Amount P10 (-0.59). This indicates
that the discrepancy between the observed and modeled frequency distributions is partly
associated with the overestimated light precipitation in models. The correlation
coefficients between the metrics other than those discussed above are lower than 0.6.
While there is some redundant information within the collection of metrics included in our
framework, we retain all metrics so that others can select an appropriate subset for their
own application. This also preserves the ability to readily identify outlier behavior of an
individual model across a wide range of regions and statistics.

4.4.    Influence of spatial resolution on metrics

Many metrics for the precipitation distribution are sensitive to the spatial resolution  of
the underlying data (e.g., Pendergrass and Knutti 2018; Chen and Dai 2019). Figure 14
shows how our results (which are all based on data at 2° resolution) are impacted if we
calculate the metrics from data coarsened to 4° grid instead. As expected, there is clearly
some sensitivity to the spatial scale at which our precipitation distribution metrics are
computed, but the correlation among datasets (both models and observations) between
the two resolutions is very high, indicating that evaluations at either resolution should be
consistent. At the coarser resolution, Amount peak and SDII are consistently smaller (as
expected); Amount P10 and Frequency P10 tend to be smaller as well. Meanwhile,
Unevenness and FracPRdays are consistently large (as expected); Amount P90,



Frequency P90, and Perkins score are generally larger as well. Chen and Dai (2019)
discussed a grid aggregation effect that is associated with the increased probability of
precipitation as the horizontal resolution becomes coarser. This effect is clearly evident
with increased Unevenness (Fig. 14g), FracPRdays (Fig. 14h), and decreased SDII (Fig.
14i) in coarser resolution. However, despite these differences, the relative model
performance is not very sensitive to the spatial scale at which we apply our analysis. The
correlation coefficients between results based on all data interpolated to 2° or 4°
horizontal resolutions are above 0.9 for all of our distribution metrics. Conclusions on
model performance are relatively insensitive to the target resolution.


**5.  Discussion**
Analyzing the distribution of precipitation intensity lags behind temperature and even
mean precipitation. Challenges include choosing appropriate metrics and analysis
resolution to characterize this highly non-gaussian variable and interpreting model skills
in the face of substantial observational uncertainty. Comparing results derived at 2° and
4° horizontal resolution for CMIP class models, we find that the quantitative changes in
assessed performance are highly consistent across models and consequently have little
impact on our conclusions. More work is needed to determine how suitable this collection
of metrics may be for evaluating models with substantially higher resolutions (e.g.,
HighResMIP, Haarsma et al. 2016). We note that more complex measures have been
designed to be scale independent (e.g., Martinez-Villalobos and Neelin 2019; Martinez-





Villalobos et al. 2022), and these may become increasingly important with continued
interest in models developed at substantially higher resolution.

Several recent studies suggest that the IMERG represents a substantial advancement
over TRMM and likely the others (e.g., Wei et al. 2017; Khodadoust Siuki et al. 2017;
Zhang et al. 2018), thus we rely on IMERG as the default in much of our analysis.
However, we do not entirely discount the other products because the discrepancy
between them provides a measure of uncertainty in the satellite-based estimates of
precipitation. Our use of the minimum to maximum range of multiple observational
products is indicative of their discrepancy, but not their uncertainty, and thus is a limitation
of the current work and challenge that we hope will be addressed in the future.

The common model biases identified in this study are mainly associated with the
overestimated light precipitation and underestimated heavy precipitation. These biases
persist from deficiencies identified in earlier generation models (e.g., Dai 2006), and as
shown in this study there has been little improvement. One reason may be that these key
characteristics of precipitation are not commonly considered in the model development
process. Enabling modelers to more readily objectively evaluate simulated precipitation
distributions could perhaps serve as a guide to improvement. The current study aims to
provide a framework for objective evaluation of simulated precipitation distributions at
regional scales.



Imperfect convective parameterizations are a possible cause of the common model
biases in precipitation distributions (e.g., Lin et al. 2013; Kooperman et al. 2018; Ahn et
al. 2018; Chen and Dai 2019; Chen et al. 2021; Martinez-Villalobos et al. 2022). Many
convective parameterizations tend to produce too frequent and light precipitation, the so-
called "drizzling" bias (e.g., Dai 2006; Trenberth et al. 2017; Chen et al. 2021; Ma et al.
2022), and it is likely due to a fact that the parameterized convection is more readily
triggered than that in the nature (e.g., Lin et al. 2013; Chen et al. 2021). As model
horizontal resolution increases, grid-scale precipitation processes can lead to resolving
convective precipitation, as in so-called cloud resolving, storm resolving, or convective
permitting models. Ma et al. (2022) compare several storm resolving models in
DYAMOND to recent CMIP6 models with a convective parameterization and observe that
the simulated precipitation distributions are more realistic in the storm resolving models.
However, some of the storm resolving models still suffer from precipitation distribution
errors, including bimodality in the frequency distribution. Further studies are needed to
better understand the precipitation distribution biases in models.


**6.   Conclusion**
We introduce a framework for regional scale evaluation of simulated precipitation
distributions with 62 climate reference regions and 10 precipitation distribution metrics
and apply it to evaluate the two most recent generations of climate model intercomparison
simulations (i.e., CMIP5 and CMIP6).





To facilitate the regional scale for evaluation, regions where precipitation characteristics
are relatively homogenous are identified. Our reference regions consist of existing IPCC
AR6 climate reference regions, with additional subdivisions based on homogeneity
analysis performed on precipitation distributions within each region. We partition the
global domain into heavy, moderate, and light precipitation regions using K-means
clustering of IMERG precipitation frequency and amount distributions. Our clustering
analysis reveals that the IPCC AR6 land regions are reasonably homogeneous in
precipitation character, while some ocean regions are relatively inhomogeneous,
including large portions of both heavy and light precipitating areas. To define more
homogeneous regions for the analysis of precipitation distributions, we have modified
some ocean regions to better fit the clustering results while retaining the original IPCC
AR6 land regions. The homogeneity between the region-averaged distribution and each
grid cell's distribution over the region is assessed by the three distinct similarity metrics
(Perkins score, K-S test, and A-D test). The homogeneity is overall improved in the
modified IPCC AR6 ocean regions. Although the clustering regions are obtained based
on the IMERG annual precipitation, the improved homogeneity is fairly consistent across
different datasets (TRMM, CMORPH, GPCP, PERSIANN, and ERA5) and seasons (MAM,
JJA, SON, and DJF). Use of these more homogeneous regions enables us to extract
more robust quantitative information from the distributions in each region.

To form the basis for evaluation within each region, we use a set of metrics that are well-
established and easy to interpret, aiming to extract key characteristics from the
distributions of daily precipitation frequency, amount, and cumulative fraction of



precipitation amount. We include the precipitation rate at the peak of the amount and
frequency distributions (Kooperman et al., 2016; Pendergrass and Deser, 2017) and
define several complementary metrics to measure the frequency and amount of
precipitation under the 10th percentile (P10) and over the 90th percentile (P90). The
distribution peak metrics assess whether the center of each distribution is shifted toward
light or heavy precipitation, while the P10 and P90 metrics quantify the fraction of light
and heavy precipitation in the distributions. The Perkins score is included to measure the
similarity between the observed and modeled frequency distributions. Also, based on the
cumulative fraction of precipitation amount, we implement the unevenness metric
counting the number of wettest days for half of the annual precipitation (Pendergrass and
Knutti 2018), the fraction of annual precipitating days above 1 mm/day, and the simple
daily intensity index (Zhang et al. 2011).

We apply the framework of regional scale precipitation distribution benchmarking to all
available realizations of 25 CMIP5 and 41 CMIP6 models and 5 satellite-based
precipitation products (IMERG, TRMM, CMORPH, GPCP, PERSIANN). The
observational discrepancy is substantially larger compared to the models' spread for
some regions, especially for mid-latitude and polar regions and for some metrics such as
Amount P90 and Frequency P90. We use two approaches to account for observational
discrepancy in the model evaluation. One is based on the number of models within the
observational range, and another is the number of models below/above all observations.
In this way, we can draw some conclusions on the overall performance in the CMIP
ensemble even in the presence of observations that may substantially disagree in certain



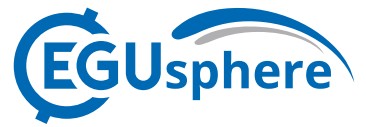

regions. Many CMIP5 and CMIP6 models underestimate the Amount and Frequency
peaks and overestimate Amount and Frequency P10 compared to observations,
especially in many mid-latitude regions where more than 50% of the models are out of
the observational range. This indicates that models produce too frequent light
precipitation, a bias that is also revealed by the overestimated FracPRdays and the
underestimated SDII. Unevenness is the metric that models simulate the worst – in many
regions more than 70-90% of the models are out of the observational range. Clear
changes in performance between CMIP5 and CMIP6 are limited. Considering all metrics,
the CMIP6 models show improvement in some mid-latitude regions, but in a few tropical
regions the CMIP6 models actually show performance degradation.

The framework presented in this study is intended to be a useful resource for model
evaluation analysts and developers working towards improved performance for a wide
range of precipitation characteristics. Basing the regions in part on homogeneous
precipitation characteristics can facilitate identification of the processes responsible for
model errors as heavy precipitating regions are generally dominated by convective
precipitation, while the moderate and light precipitation regions are mainly governed by
stratiform precipitation processes. Although the framework presented herein has been
demonstrated with regional scale evaluation benchmarking, it can be applicable for
benchmarking at larger scales and homogeneous precipitation regions.





**Code Availability**

The benchmarking framework for precipitation distributions established in this study is available via the PCMDI Metrics Package (PMP, https://github.com/PCMDI/pcmdi_metrics, DOI: 10.5281/zenodo.7231033). This framework provides three tiers of area averaged outputs for i) large scale domain (Tropics and Extratropics with separated land and ocean) commonly used in the PMP, ii) large scale domain with clustered precipitation characteristics (Tropics and Extratropics with separated land and ocean, and separated heavy, moderate, and light precipitation regions), and iii) modified IPCC AR6 regions shown in this paper.

**Data Availability**

All of the data used in this study are publicly available. The satellite-based precipitation products used in this study (IMERG, TRMM, CMORPH, GPCP, and PERSIANN) and ERA5 precipitation product are available on the Obs4MIPs at https://esgf-node.llnl.gov/projects/obs4mips/. The CMIP data is available on the ESGF at https://esgf-node.llnl.gov/projects/esgf-llnl.

**Author contribution**

PG and AP designed the initial idea of the precipitation benchmarking framework. MA, PU, PG, and JL advanced the idea and developed the framework. MA performed analysis.



MA, JL, and AO implemented the framework code into the PCMDI metrics package. MA
prepared the manuscript with contributions from all co-authors.


**Competing interests**
The authors declare that they have no conflict of interest.


**Disclaimer**
This document was prepared as an account of work sponsored by an agency of the U.S.
government. Neither the U.S. government nor Lawrence Livermore National Security,
LLC, nor any of their employees makes any warranty, expressed or implied, or assumes
any legal liability or responsibility for the accuracy, completeness, or usefulness of any
information, apparatus, product, or process disclosed, or represents that its use would
not infringe privately owned rights. Reference herein to any specific commercial product,
process, or service by trade name, trademark, manufacturer, or otherwise does not
necessarily constitute or imply its endorsement, recommendation, or favoring by the U.S.
government or Lawrence Livermore National Security, LLC. The views and opinions of
authors expressed herein do not necessarily state or reflect those of the U.S. government
or Lawrence Livermore National Security, LLC, and shall not be used for advertising or
product endorsement purposes.






**Acknowledgements**
This work was performed under the auspices of the U.S. Department of Energy by
Lawrence Livermore National Laboratory under Contract DE-AC52-07NA27344. The
efforts of the authors were supported by the Regional and Global Model Analysis (RGMA)
program of the United States Department of Energy's Office of Science, including under
Award Number DE-SC0022070 and National Science Foundation (NSF) IA 1947282.
This work was also partially supported by the National Center for Atmospheric Research
(NCAR), which is a major facility sponsored by the NSF under Cooperative Agreement
No. 1852977. We acknowledge the World Climate Research Programme's Working
Group on Coupled Modeling, which is responsible for CMIP, and we thank the climate
modeling groups for producing and making available their model output, the Earth System
Grid Federation (ESGF) for archiving the output and providing access, and the multiple
funding agencies who support CMIP and ESGF. The U.S. Department of Energy's
Program for Climate Model Diagnosis and Intercomparison (PCMDI) provides
coordinating support and led development of software infrastructure for CMIP.




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





**Tables**

Table 1. Satellite-based and reanalysis precipitation products used in this study.

| Product | Data source | Coverage | | Resolution | | Reference |
|---|---|---|---|---|---|---|
| | | Domain | Period | Horizontal | Frequency | |
| IMERG | NASA Integrated Multi-satellitE Retrievals for GPM version 6 final run product | Global, while beyond 60°NS is incomplete | 2000.6-present | 0.1° | 30 minutes | Huffman et al. (2020) |
| TRMM | NASA Tropical Rainfall Measuring Mission Multi-satellite Precipitation Analysis 3B42 version 7 product | 50°S-50°N | 1998.1-2019.12 | 0.25° | 3 hours | Huffman et al. (2007) |
| CMORPH | NOAA Bias-corrected Climate Prediction Center Morphing technique product | 60°S-60°N | 1998.1-present | 0.073° | 30 minutes | Xie et al. (2017) |
| GPCP | NASA Global Precipitation Climatology Project 1DD version 1.3 | Global, while beyond 40°NS is incomplete | 1996.10-present | 1° | 1 day | Huffman et al. (2001) |
| PERSIANN | UC-IRVINE/CHRS Precipitation Estimation from Remotely Sensed Information using Artificial Neural Networks-Climate Data Record | 60°S-60°N | 1983.1-present | 0.25° | 1 day | Ashouri et al. (2015) |
| ERA5 | ECMWF Integrated Forecasting System Cy41r2 | Global | 1950.1–present | 0.25° | 1 hour | Hersbach et al. (2020) |

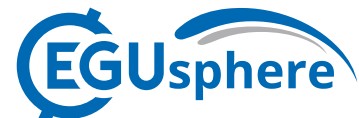

910

Table 2. CMIP5 and CMIP6 models used in this study and their horizontal resolution. The number in parentheses indicates the number of realizations used for each model. Note that the horizontal resolution information is obtained from the number of grids, and it may vary slightly if the grid interval is not linear.


| Institute | CMIP5 | | CMIP6 | |
|---|---|---|---|---|
| | Name | Horizontal resolution [lon x lat °] | Name | Horizontal resolution [lon x lat °] |
| CSIRO/BOM, Australia | ACCESS1-0 (1) | 1.875 x 1.241 | ACCESS-CM2 (7) | 1.875 x 1.25 |
| | ACCESS1-3 (2) | 1.875 x 1.241 | ACCESS-ESM1-5 (10) | 1.875 x 1.241 |
| BCC, China | BCC-CSM1-1 (3) | 1.875 x 1.241 | BCC-CSM2-MR (3) | 1.125 x 1.125 |
| | BCC-CSM1-1-M (3) | 1.125 x 1.125 | BCC-ESM1 (3) | 2.812 x 2.812 |
| BNU, China | BNU-ESM (1) | 2.812 x 2.812 | N/A | |
| CAMS, China | N/A | | CAMS-CSM1-0 (3) | |
| CCCma, Canada | N/A | | CanESM5 (7) | 2.812 x 2.812 |
| NCAR, USA | CCSM4 (6) | 1.25 x 0.938 | CESM2 (10) | 1.25 x 0.938 |
| | | | CESM2-FV2 (3) | 2.5 x 1.875 |
| | | | CESM2-WACCM (3) | 1.25 x 0.938 |
| | | | CESM2-WACCM-FV2 (3) | 2.5 x 1.875 |
| CMCC, Italy | CMCC-CM (3) | 0.75 x 0.75 | CMCC-CM2-HR4 (1) | 1.25 x 0.938 |
| | | | CMCC-CM2-SR5 (1) | 1.25 x 0.938 |
| CNRM-CERFACS, France | N/A | | CNRM-CM6-1 (1) | 1.406 x 1.406 |
| | | | CNRM-CM6-1-HR (1) | 0.5 x 0.5 |
| | | | CNRM-ESM2-1 (1) | 1.406 x 1.406 |
| CSIRO-QCCCE, Australia | CSIRO-Mk3-6-0 (10) | 1.875 x 1.875 | N/A | |
| DOE, USA | N/A | | E3SM-1-0 (3) | 1.0 x 1.0 |
| EC-Earth-Consortium, European Community | EC-Earth (1) | 1.125 x 1.125 | EC-Earth3 (6) | 0.703 x 0.703 |
| | | | EC-Earth3-AerChem (1) | 0.703 x 0.703 |
| | | | EC-Earth3-CC (5) | |
| | | | EC-Earth3-Veg (3) | 0.703 x 0.703 |
| IAP-CAS/THU, China | FGOALS-g2 (1) | 2.812 x 3.0 | FGOALS-f3-L (3) | 1.0 x 1.0 |
| | FGOALS-s2 (3) | 2.812 x 1.667 | | |
| NOAA GFDL, USA | GFDL-CM3 (5) | 2.5 x 2.0 | GFDL-CM4 (1) | 1.0 x 1.0 |
| | GFDL-HIRAM-C180 (2) | 0.625 x 0.5 | GFDL-ESM4 (1) | 1.0 x 1.0 |
| | GFDL-HIRAM-C360 (1) | 0.312 x 0.25 | | |
| NASA GISS, USA | GISS-E2-R (2) | 2.5 x 2.0 | N/A | |
| MOHC, UK | HadGEM2-A (1) | 1.875 x 1.241 | HadGEM3-GC31-LL (5) | 1.875 x 1.25 |
| | | | HadGEM3-GC31-MM (4) | 0.833 x 0.556 |
| | | | UKESM1-0-LL (1) | 1.875 x 1.25 |
| IITM, India | N/A | | IITM-ESM (1) | 1.875 x 1.915 |
| INM, Russia | INMCM4 (1) | 2.0 x 1.5 | INM-CM4-8 (1) | 2.0 x 1.5 |
| | | | INM-CM5-0 (1) | 2.0 x 1.5 |



| | | | | |
|---|---|---|---|---|
| IPSL, France | IPSL-CM5A-LR (6) | 3.75 x 1.875 | IPSL-CM6A-LR (22) | 2.5 x 1.259 |
| | IPSL-CM5A-MR (3) | 2.5 x 1.259 | | |
| | IPSL-CM5B-LR (1) | 3.75 x 1.875 | | |
| NIMS/KMA, Korea | N/A | | KACE-1-0-G (1) | 1.875 x 1.25 |
| MIROC, Japan | MIROC5 (2) | 1.406 x 1.406 | MIROC6 (10) | 1.406 x 1.406 |
| | | | MIROC-ES2L (3) | 2.812 x 2.812 |
| MPI-M, Germany | MPI-ESM-MR (3) | 1.875 x 1.875 | MPI-ESM-1-2-HAM (3) | 1.875 x 1.875 |
| | | | MPI-ESM1-2-HR (3) | 0.938 x 0.938 |
| | | | MPI-ESM1-2-LR (3) | 1.875 x 1.875 |
| MRI, Japan | MRI-AGCM3-2H (1) | 0.562 x 0.562 | MRI-ESM2-0 (3) | 1.125 x 1.125 |
| | MRI-AGCM3-2S (1) | 0.188 x 0.188 | | |
| | MRI-CGCM3 (3) | 1.125 x 1.125 | | |
| NCC, Norway | N/A | | NorCPM1 (10) | 2.5 x 1.875 |
| | | | NorESM2-LM (2) | 2.5 x 1.875 |
| SNU, Korea | N/A | | SAM0-UNICON (1) | 1.25 x 0.938 |
| AS-RCEC, Taiwan | N/A | | TaiESM1 (1) | 1.25 x 0.938 |







Table 3. Precipitation distribution metrics implemented in this study.

| Metric [unit] | Definition | Objectives | Reference |
|---|---|---|---|
| **Amount peak** [mm/day] | Rain rate where the maximum rain amount occurs | Characterize typical daily precipitation amount | Pendergrass and Deser (2017) |
| **Amount P10** [fraction] | Fraction of rain amount in lower 10 percentile of OBS amount | Measure the rain amount from light rainfall | |
| **Amount P90** [fraction] | Fraction of rain amount in upper 90 percentile of OBS amount | Measure the rain amount from heavy rainfall | |
| **Frequency peak** [mm/day] | Rain rate where the maximum nonzero rain frequency occurs | Characterize typical daily precipitation frequency | Pendergrass and Deser (2017) |
| **Frequency P10** [fraction] | Fraction of rain frequency in lower 10 percentile of OBS amount | Measure the frequency of light rainfall | |
| **Frequency P90** [fraction] | Fraction of rain frequency in upper 90 percentile of OBS amount | Measure the frequency of heavy rainfall | |
| **Unevenness** [days] | Number of the wettest days for that constitute half of annual precipitation | Measure uneven characteristic of daily precipitation | Pendergrass and Knutti (2018) |
| **FracPRdays** [fraction] | Number of precipitating days (>=1mm/day) divided by total days a year | Measure fraction of precipitating days a year | |
| **SDII** [mm/day] | Annual total precipitation divided by the number of precipitating days (>=1mm/day) | Measure daily precipitation intensity | Zhang et al. (2011) |
| **Perkins score** [unitless between 0-1] | Sum of minimum values between two PDFs across all bins | Measure similarity between two PDFs | Perkins et al. (2007) |







Table 4. List of climate reference regions used in this study. The new ocean regions
defined in this study are highlighted in bold.

| 1 | GIC | Greenland/Iceland | 22 | WAF | Western-Africa | 43 | SAU | S.Australia |
|---|-----|-------------------|----|-----|----------------|----|-----|-------------|
| 2 | NWN | N.W.North-America | 23 | CAF | Central-Africa | 44 | NZ | New-Zealand |
| 3 | NEN | N.E.North-America | 24 | NEAF | N.Eastern-Africa | 45 | EAN | E.Antarctica |
| 4 | WNA | W.North-America | 25 | SEAF | S.Eastern-Africa | 46 | WAN | W.Antarctica |
| 5 | CNA | C.North-America | 26 | WSAF | W.Southern-Africa | 47 | ARO | Arctic-Ocean |
| 6 | ENA | E.North-America | 27 | ESAF | E.Southern-Africa | 48 | ARS | Arabian-Sea |
| 7 | NCA | N.Central-America | 28 | MDG | Madagascar | 49 | BOB | Bay-of-Bengal |
| 8 | SCA | S.Central-America | 29 | RAR | Russian-Arctic | 50 | EIO | Equatorial-Indian-Ocean |
| 9 | CAR | Caribbean | 30 | WSB | W.Siberia | 51 | SIO | S.Indian-Ocean |
| 10 | NWS | N.W.South-America | 31 | ESB | E.Siberia | **52** | **NPO** | **N.Pacific-Ocean** |
| 11 | NSA | N.South-America | 32 | RFE | Russian-Far-East | **53** | **NWPO** | **N.W.Pacific-Ocean** |
| 12 | NES | N.E.South-America | 33 | WCA | W.C.Asia | **54** | **NEPO** | **N.E.Pacific-Ocean** |
| 13 | SAM | South-American-Monsoon | 34 | ECA | E.C.Asia | **55** | **PITCZ** | **Pacific-ITCZ** |
| 14 | SWS | S.W.South-America | 35 | TIB | Tibetan-Plateau | **56** | **SWPO** | **S.W.Pacific-Ocean** |
| 15 | SES | S.E.South-America | 36 | EAS | E.Asia | **57** | **SEPO** | **S.E.Pacific-Ocean** |
| 16 | SSA | S.South-America | 37 | ARP | Arabian-Peninsula | **58** | **NAO** | **N.Atlantic-Ocean** |
| 17 | NEU | N.Europe | 38 | SAS | S.Asia | **59** | **NEAO** | **N.E.Atlantic-Ocean** |
| 18 | WCE | West&Central-Europe | 39 | SEA | S.E.Asia | **60** | **AITCZ** | **Atlantic-ITCZ** |
| 19 | EEU | E.Europe | 40 | NAU | N.Australia | **61** | **SAO** | **S.Atlantic-Ocean** |
| 20 | MED | Mediterranean | 41 | CAU | C.Australia | **62** | **SOO** | **Southern-Ocean** |
| 21 | SAH | Sahara | 42 | EAU | E.Australia | | | |






**Figures**

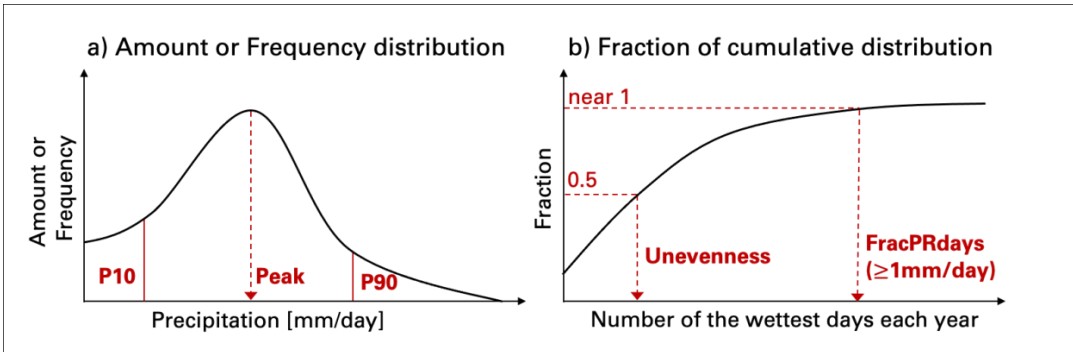

Figure 1. Schematics for precipitation distribution metrics. a) Amount or Frequency distribution as a function of rain rate. Peak metric gauges the rain rate where the maximum distribution occurs. P10 and P90 metrics respectively measure the fraction of the distribution lower 10 percentile and upper 90 percentile. Perkins score is another metric based on the frequency distribution to quantify the similarity between observed and modeled distribution. b) Fraction of cumulative distribution as a function of number of the wettest days. Unevenness gauges the number of the wettest days for half of annual precipitation. FracPRdays measures the fraction of the number of precipitating (≥1mm/day) days a year. SDII is designed to measure daily precipitation intensity by annual total precipitation divided by FracPRdays.






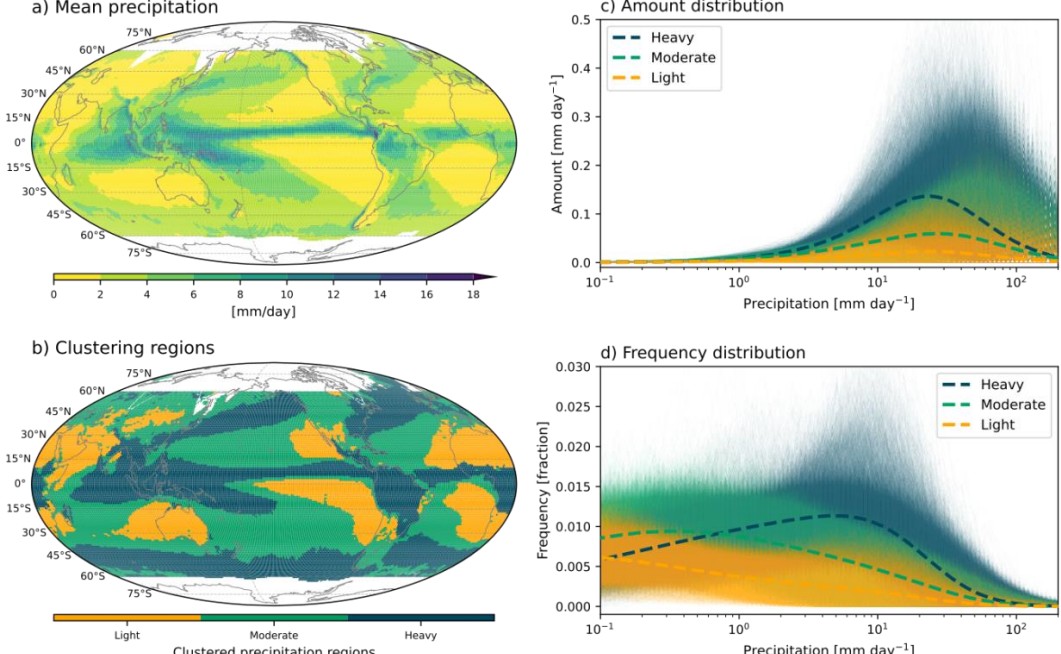


Figure 2. Spatial patterns of IMERG precipitation a) mean state and b) clustering for
heavy, moderate, and light precipitating regions by K-means clustering with amount and
frequency distributions. Precipitation c) amount and d) frequency distributions as a
function of rain rate. Different colors indicate different clustering regions as the same
with b). Thin and thick curves respectively indicate distributions at each grid and the
cluster average.



Figure 3. a) IPCC AR6 climate reference regions and b) modified IPCC AR6 climate reference regions superimposed on the precipitation distributions clustering map shown in Fig. 2b. Land regions are the same between a) and b), while some ocean regions are modified.




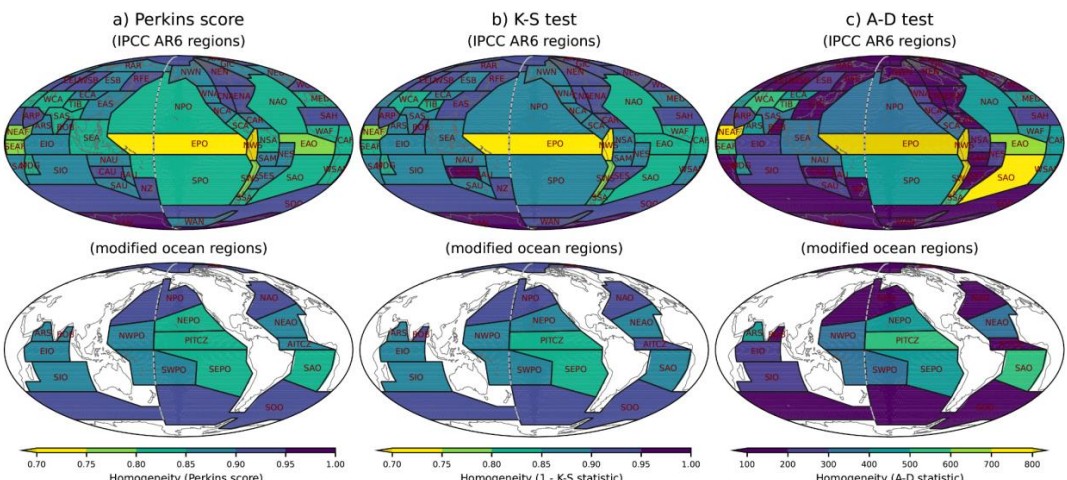


Figure 4. Homogeneity estimated by a) Perkins score, b) K-S test, and c) A-D test
between the region averaged and each grid's frequency distributions of IMERG
precipitation for the IPCC AR6 climate reference regions (upper) and the modified
ocean regions (bottom). Darker color indicates higher homogeneity across all panels.



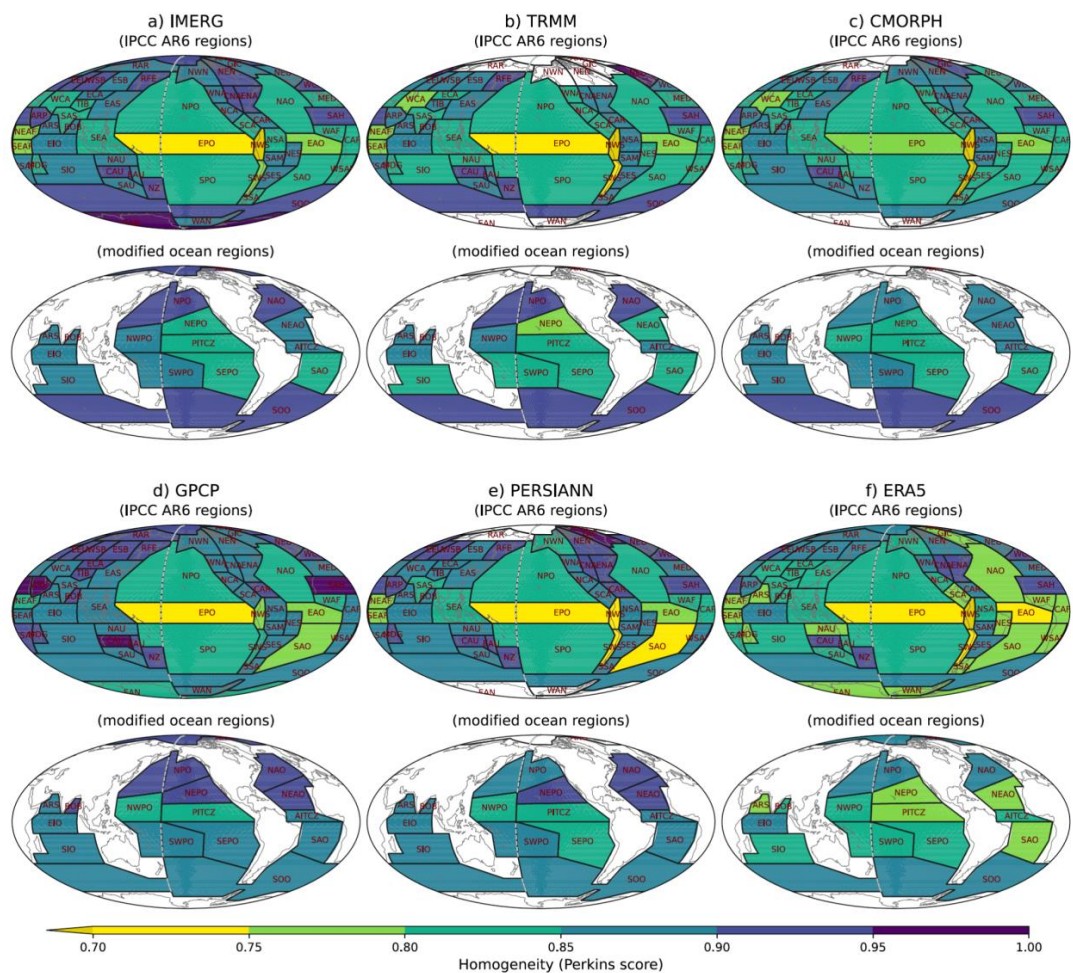

Figure 5. As in Fig. 4, but for different observational datasets with Perkins score.






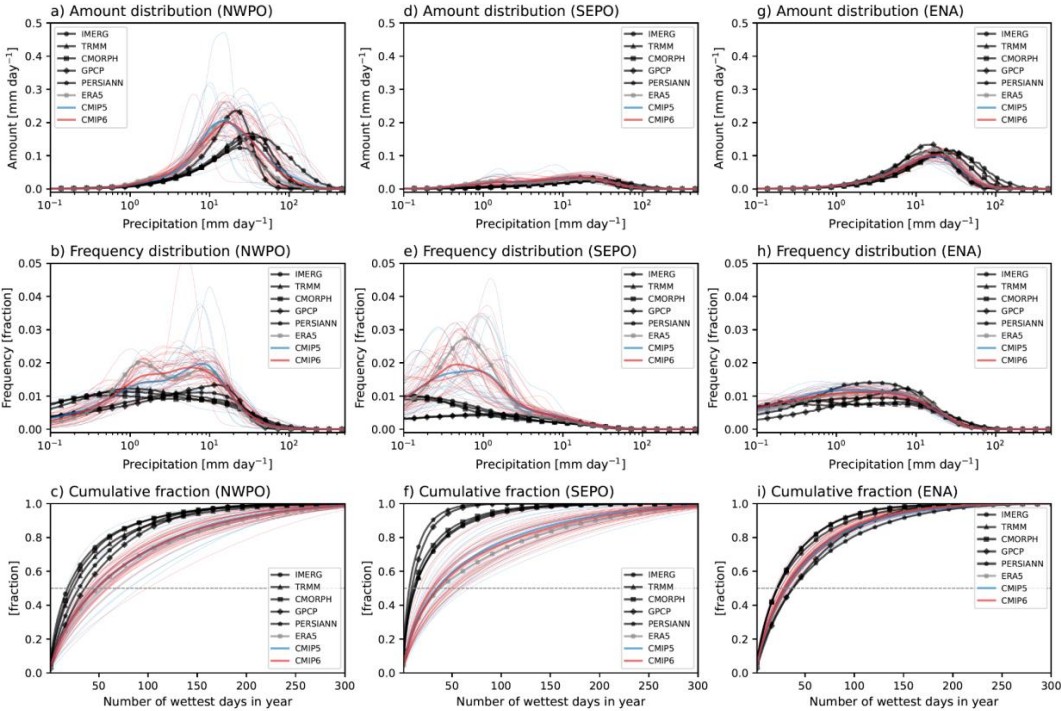

Figure 6. Precipitation amount (upper), frequency (middle), and cumulative (bottom)
distributions for a-c) NWPO, b-f) SEPO, and g-j) ENA. Black, gray, blue, and red curves
indicate the satellite-based observations, reanalysis, CMIP5 models, and CMIP6
modes, respectively. Thin and thick curves for CMIP models respectively indicate
distributions for each model and multi-model average. Gray dotted lines in the
cumulative distributions indicate a fraction of 0.5. Note: all model output and
observations were conservatively regridded to 2° in the first step of analysis.




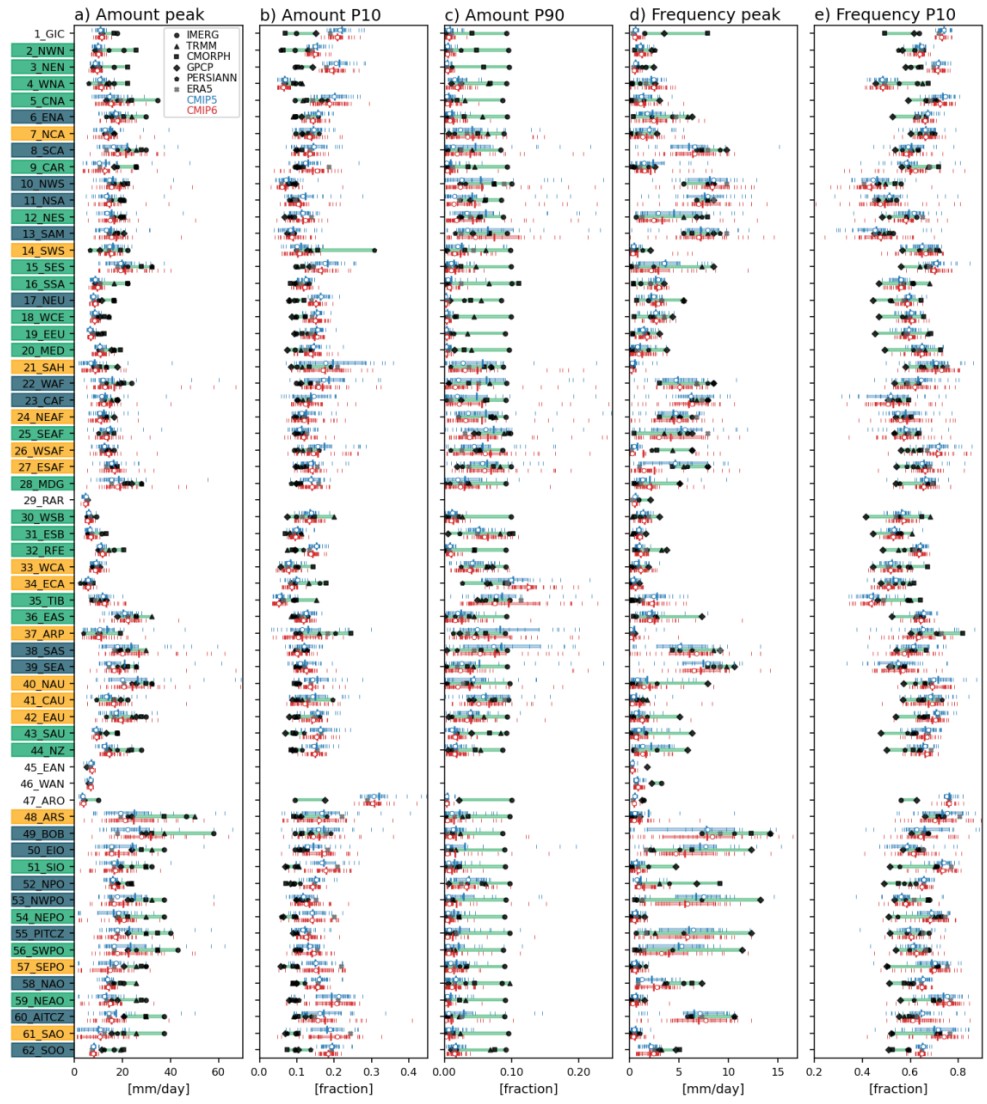

Figure 7. Precipitation distribution metrics for a) Amount peak, b) Amount P10, c)
Amount P90, d) Frequency peak, e) Frequency P10, f) Frequency P90, g) Unevenness,
h) FracPRdays, i) SDII, and j) Perkins score over the modified IPCC AR6 regions.
Black, gray, blue, and red curves indicate the satellite-based observations, reanalysis,
CMIP5 models, and CMIP6 modes, respectively. Thin and thick vertical marks for CMIP
models respectively indicate distributions for each model and multi-model average.
Open circle mark for CMIP models indicates the multi-model median. Green shade





represents the range between the minimum and maximum values of satellite-based
observations. Blue and red shades respectively represent the range between 25th and
75th model values for CMIP 5 and 6 models. Y-axis labels are shaded with the three
colors as the same in Fig. 2b, indicating dominant precipitating characteristics. Note that
regions 1-46 are land and land-ocean mixed regions, and 47-62 are ocean regions.

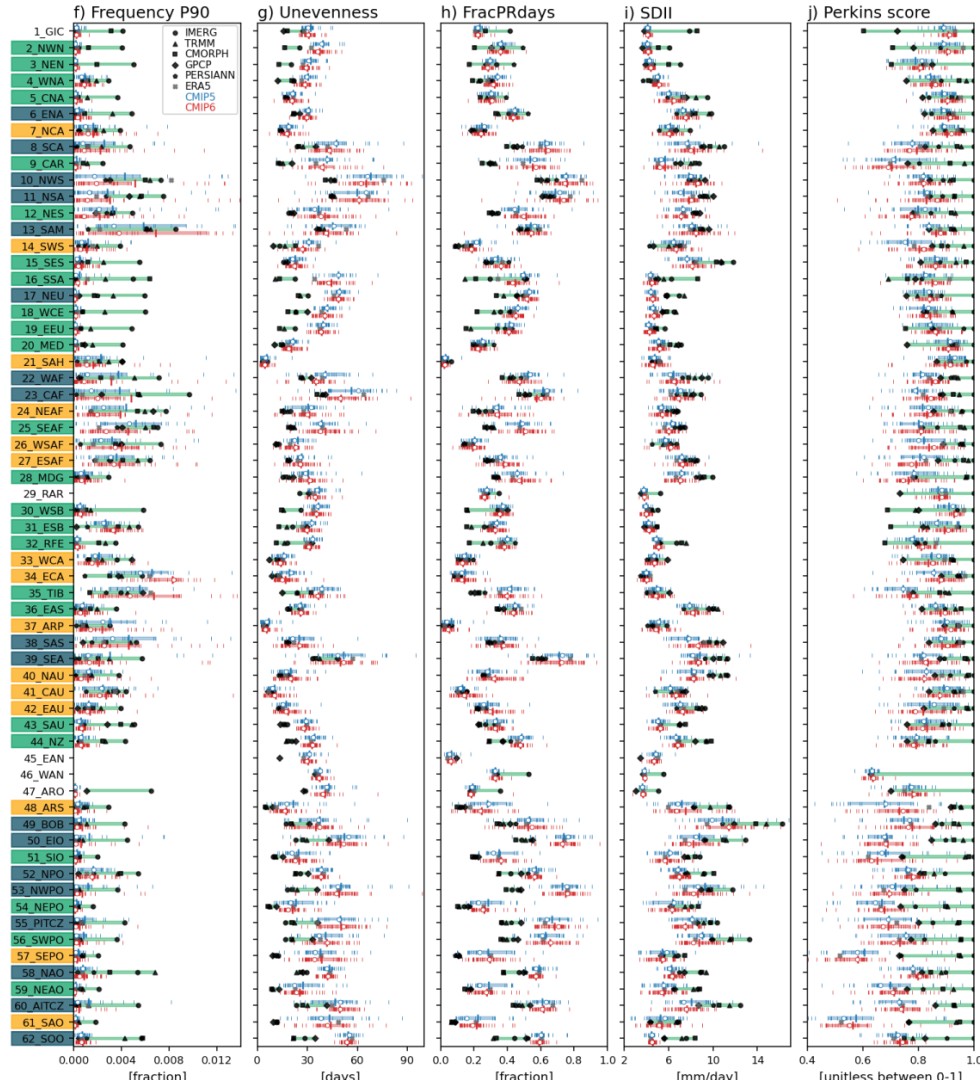

Figure 7. (continued)




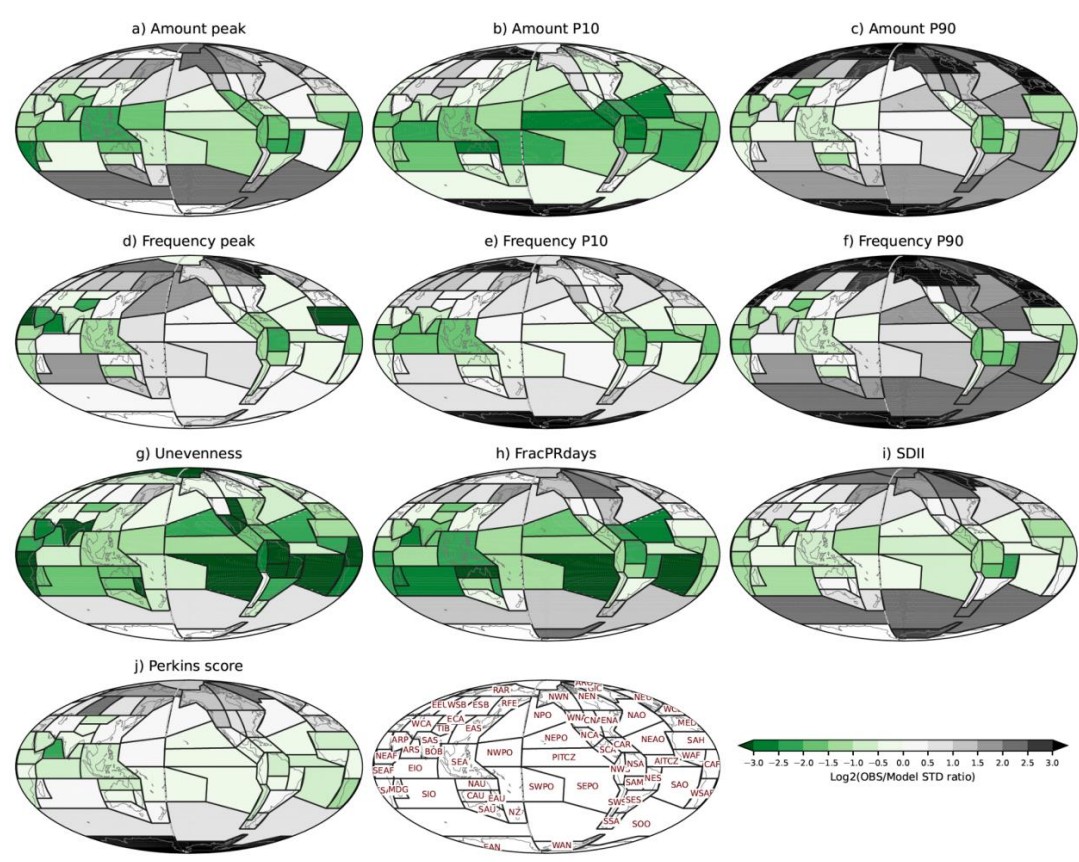


Figure 8. Observational discrepancies relative to spread in the multi-model ensemble for
a) Amount peak, b) Amount P10, c) Amount P90, d) Frequency peak, e) Frequency
P10, f) Frequency P90, g) Unevenness, h) FracPRdays, i) SDII, and j) Perkins score
over the modified IPCC AR6 regions. The observational discrepancy is calculated by
the standard deviation of satellite-based observations divided by the standard deviation
of CMIP 5 and 6 models for each metric and region.





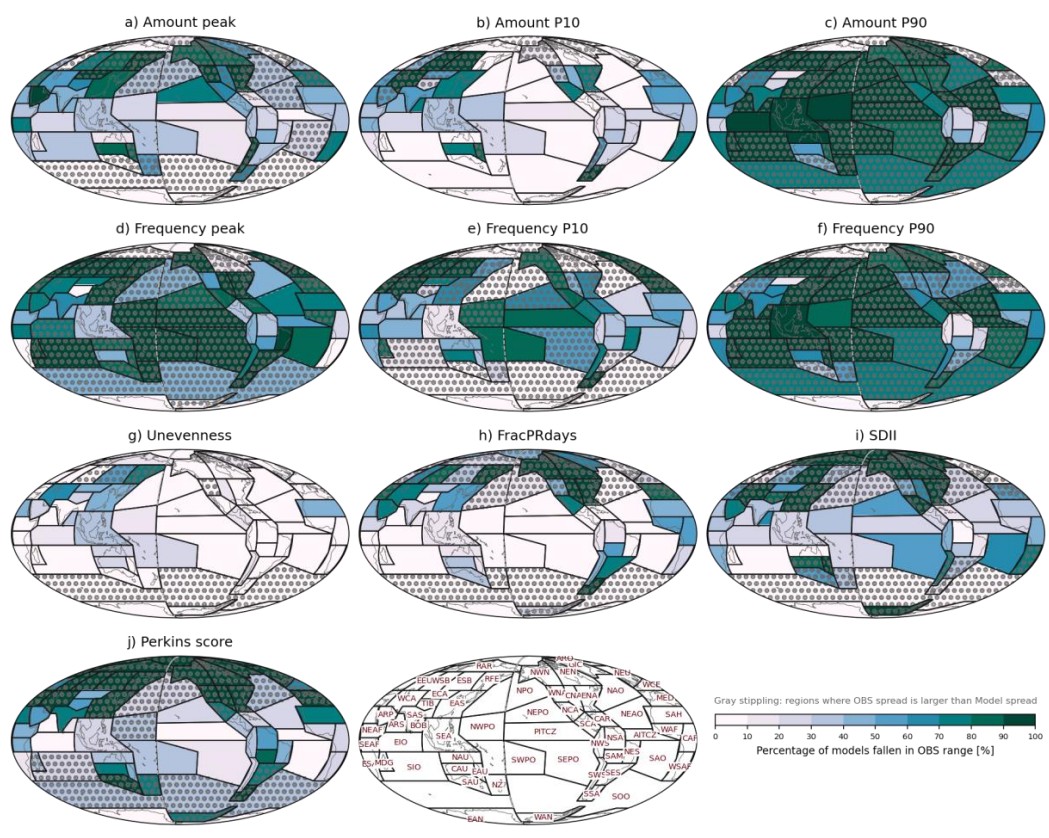

Figure 9. Percentage of CMIP6 models within range of the observational products for a)
Amount peak, b) Amount P10, c) Amount P90, d) Frequency peak, e) Frequency P10, f)
Frequency P90, g) Unevenness, h) FracPRdays, i) SDII, and j) Perkins score over the
modified IPCC AR6 regions. The observational range is between the minimum and
maximum values of five satellite-based products. Regions where the observational
spread is larger than model spread shown in Fig. 8 are stippled gray.






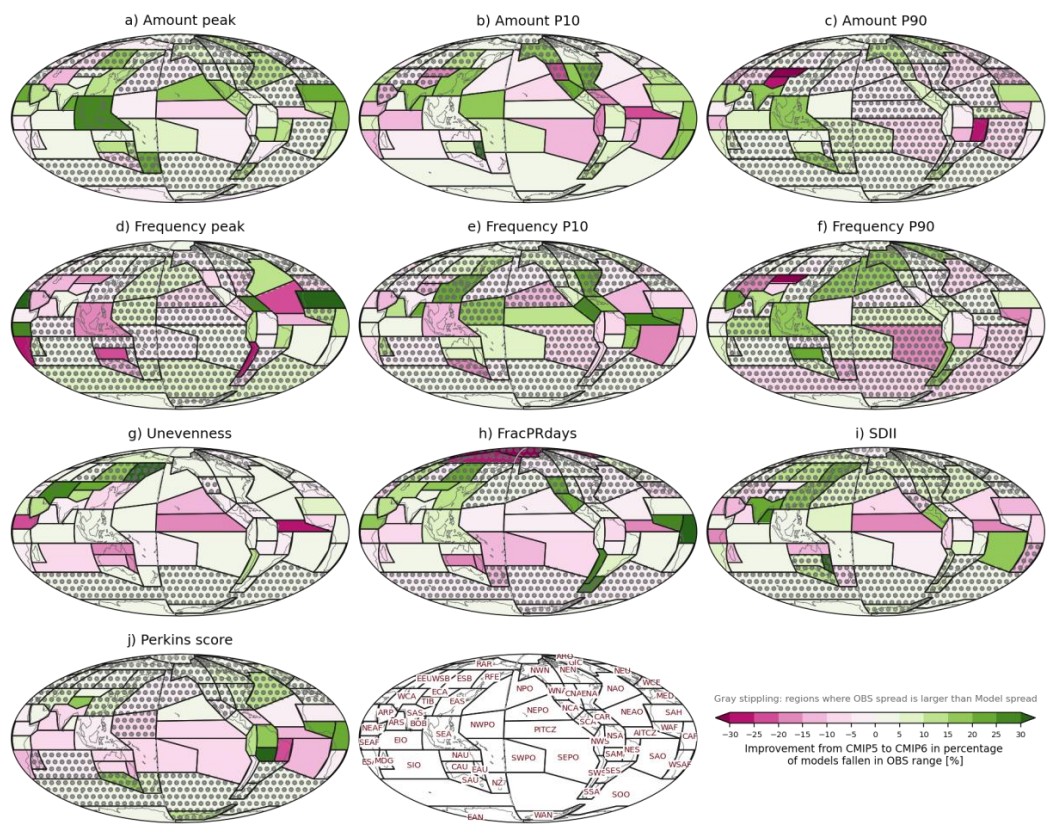


Figure 10. Improvement from CMIP 5 to 6 as identified by the percentage of models in
each multi-model ensemble that are within the observational min-to-max range. The
improvement is calculated by the CMIP6 percentage minus the CMIP5 percentage, so
that positive and negative values respectively indicate improvement and deterioration in
CMIP6. Regions where the observational spread is larger than model spread are
stippled gray.



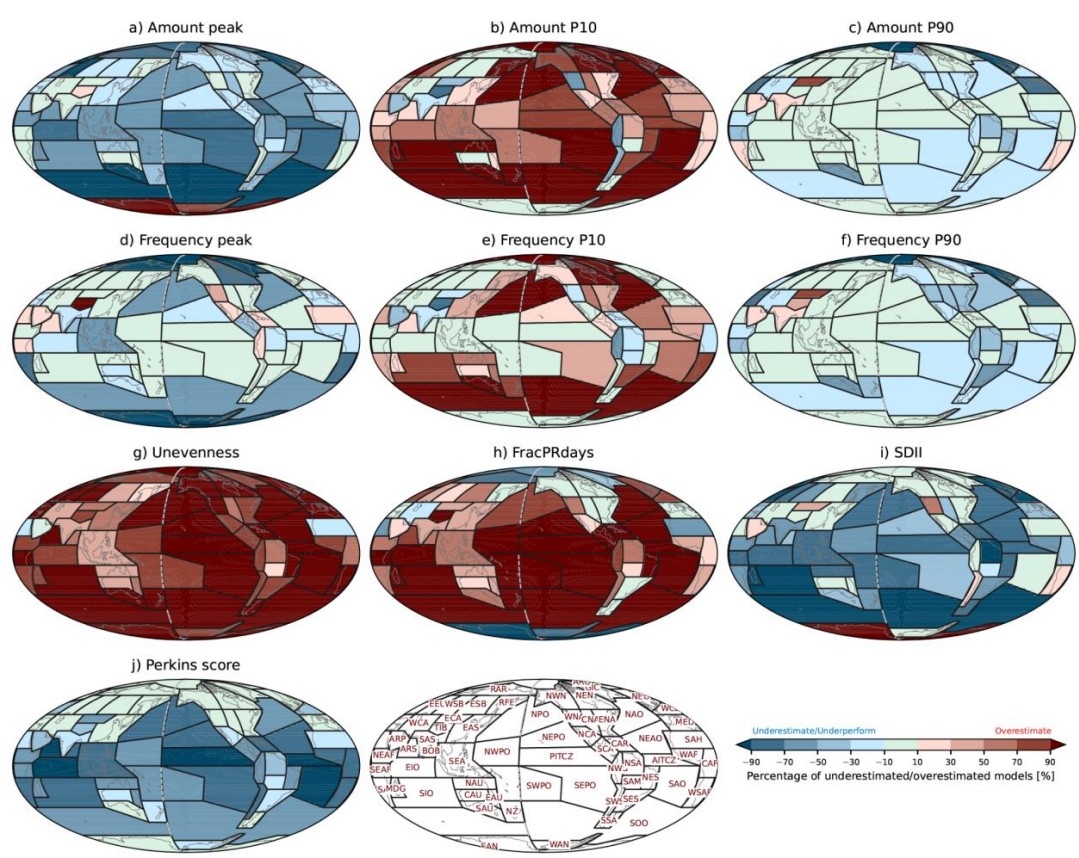

Figure 11. Percentage of CMIP6 models underestimating or overestimating observations for a) Amount peak, b) Amount P10, c) Amount P90, d) Frequency peak, e) Frequency P10, f) Frequency P90, g) Unevenness, h) FracPRdays, i) SDII, and j) Perkins score over the modified IPCC AR6 regions. The criteria for underestimation and overestimation are respectively defined by minimum and maximum values of satellite-based observations shown in Fig. 7. Positive and negative values respectively represent overestimation and underestimation by a formulation of $(nO - nU)/nT$ where $nO, nU, nT$ are respectively the number of overestimated models, underestimated models, and total models.






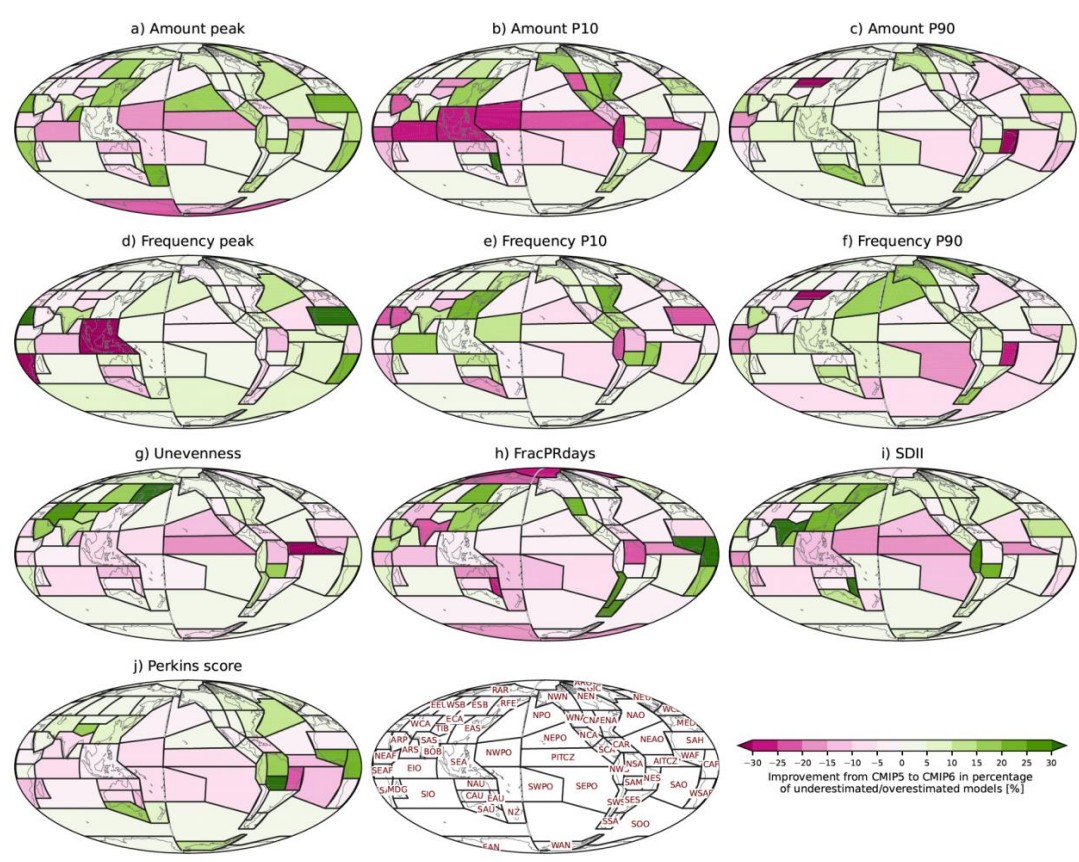


Figure 12. Improvement from CMIP 5 to 6 in the percentage of underestimated or overestimated models. The improvement is calculated by the absolute value of CMIP5 percentage minus the absolute value of CMIP6 percentage, so that positive and negative values respectively indicate improvement and deterioration in CMIP6.





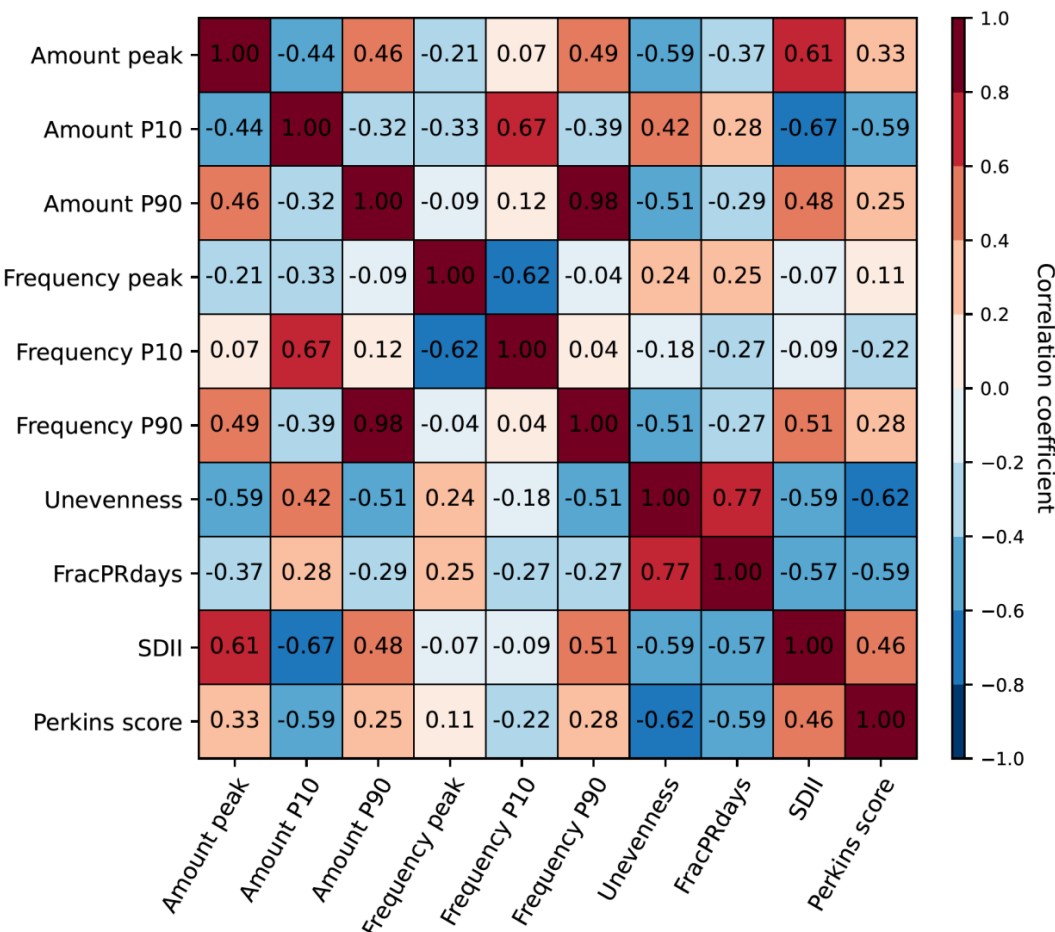

Figure 13. Correlation between precipitation distribution metrics across CMIP 5 and 6
model performances. The correlation coefficients are calculated for the modified IPCC
AR6 regions and then area-weighted averaged globally.



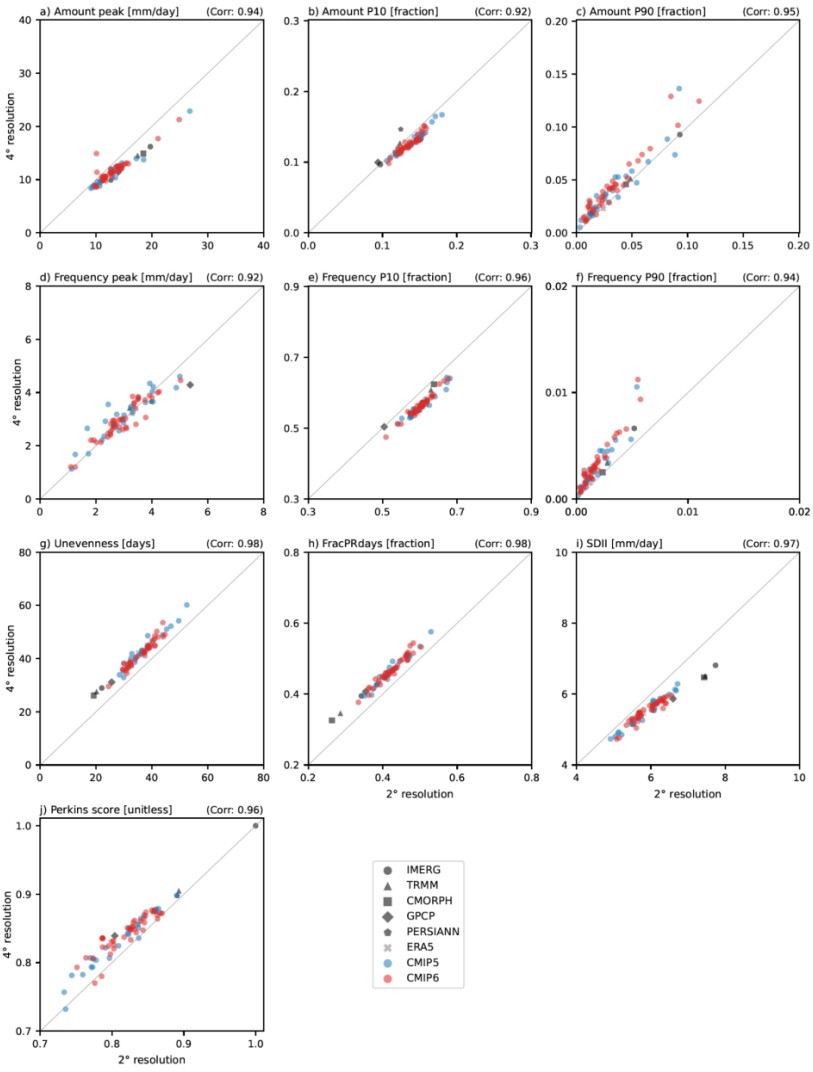

Figure 14. Scatterplot between 2° and 4° interpolated horizontal resolutions in
evaluating precipitation distribution metrics for a) Amount peak, b) Amount P10, c)
Amount P90, d) Frequency peak, e) Frequency P10, f) Frequency P90, g) Unevenness,
h) FracPRdays, i) SDII, and j) Perkins score. The metric values are calculated for the
modified IPCC AR6 regions and then weighted averaged globally. Black, gray, blue, and
red marks indicate the satellite-based observations, reanalysis, CMIP5 models, and
CMIP6 modes, respectively. The number in the upper right of each panel is the
correlation coefficient between the metric values in 2° and 4° resolutions across all
observations and models.