# Peer review of "Evaluating Precipitation Distributions at Regional Scales: A Benchmarking Framework and Application to CMIP 5 and 6 Models"

_EGUsphere, 2022_

## Author Response (AR1)

**Referee comments #1**

Major comments

This manuscript presents a framework for quantifying precipitation distributions at regionsl scales, which is interesting to the readers of the GMD and the community. If this is applied to CMIP5 and CMIP6, they found a few interesting results: (1) overall overestimation of light precipitation, (2) improvement in mid-latitude regions, and (3) degradation over the tripics. Overall, the authors did good job in listing details of the analysis method and result.

We thank the reviewer for the helpful and constructive comments. Please find our point-by-point response below.

1. The intent of the authors seems quite clear. However, I do have impression that kwown/widely applied metrics are just applied over smaller regions - based on the CMIP6 regions. It is not clear to me what are the novelity of this manuscript.

We think our manuscript has novelties in addition to the evaluation of simulated precipitation at regional scales. In this study we brought together a diverse suite of well-established precipitation distribution metrics and several new complementary metrics, providing a more comprehensive objective assessment of precipitation distributions than earlier studies that generally rely on only one or two metrics. This more comprehensive objective perspective has enabled us to assess which metrics are most informative, and which regions of the world have similar bias characteristics across CMIP models. In addition to that, via cluster analysis we propose a new suite of oceanic reference regions for evaluating simulated precipitation that are more homogeneous in precipitation character than those used in the IPCC AR6.

To clarify these points in the manuscript, we have revised the last paragraph of the introduction section. Below is the revised sentence:

"In this study, we propose a modified IPCC AR6 reference regions and a framework for regional scale quantification of simulated precipitation distributions, which is implemented into the PCMDI metrics package to enable researchers to readily use the metric collection in a common framework."

2. Defining homogeneous region can also be subjective depending on the criteria chosen.

We agree with the reviewer that generally defining homogeneous regions is subjective depending on the criteria chosen. However, we did not use any criteria of precipitation for defining the homogeneous regions, rather, it is one of the outcomes of the K-means clustering. The K-means clustering algorithm defines the homogeneity regions by iterating to minimize the sum of distances

between the cluster centroid and each cluster member. The only parameter for K-means clustering is the number of desired clusters. Here we use 3 for defining heavy, moderate, and light rain regions.

We have added more information about this in the revised manuscript. Below are the added/revised sentences in the method section:

"K-means clustering is an unsupervised machine learning algorithm that separates characteristics of a dataset into a given number of clusters without explicitly provided criteria. This method has been widely used because it is faster and simpler than other methods. Here, we use 3 clusters to define heavy, moderate, and light precipitation regions."

Overall, I do feel this well-done research and is worth to be published. I don't think one can have clear answer on #1 or #2, but do hope to see more clear scope and goals of the manuscript.

We thank the reviewer's comments that help improve our manuscript.

**Referee comments #2**

The paper provides a thorough model intercomparison as to how CMIP5 and 6 models simulate different characteristics of precipitation distributions across IPCC AR6 subregions and modified ocean regions. Across all metrics and globally, they note limited improvement in performance between CMIP5 and CMIP6, confined to some mid-latitude regions. Overall, the authors were thorough and clear in their justification and selection of metrics. I particularly like Figure 1 and the authors' methods to quantify observational discrepancy as well.

We thank the reviewer for the helpful and constructive comments. Please find our point-by-point response below.

My main suggestions and comments are as follows:

1.) After reading the full manuscript, I am not exactly sure what the authors' 'Benchmarking Framework' is or how I could use it in my research, other than the consolidated metrics and possibly the methods for quantifying observational discrepancy. "Evaluation" and "Benchmarking" are used synonymously, despite distinct differences (see Abramowitz 2005, 2012). I am unsure what the benchmark is, or if the authors are establishing the benchmark for the next generation of CMIP. This could be clarified and stated explicitly within or following the second paragraph of the introduction and ideally in the abstract as well. It might also be beneficial to the reader to have the components of the framework summarized explicitly in the intro or discussion section.

We appreciate the reviewer's comment, which has helped us to clarify our benchmarking framework. As suggested, we have added more information about benchmarking in the second paragraph of the introduction section. Below are the added/revised sentences in the revised manuscript:

"As discussed in previous studies (e.g., Abramowitz 2012), our reference to model benchmarking implies model evaluation with community-established reference data sets, performance tests (metrics), variables, and spatial and temporal resolutions. The U.S. Department of Energy (DOE) envisioned a framework for both baseline and exploratory precipitation benchmarks (U.S. DOE. 2020) as summarized by Pendergrass et al. (2020). While the exploratory benchmarks focus on process-oriented and phenomena-based metrics at a variety of spatiotemporal scales (Leung et al. 2022), the baseline benchmarks target well-established measures such as mean state, the seasonal and diurnal cycles, variability across timescales, intensity/frequency distributions, extremes, and drought (e.g., Gleckler et al. 2008; Covey et al. 2016; Wehner et al. 2020; Ahn et al. 2022). The current study builds on the baseline benchmarks by proposing a framework for benchmarking simulated precipitation distributions against multiple observations using well-established metrics and reference regions. To ensure consistent application of this framework, the metrics used herein are implemented and made available as part of the widely-used Program for Climate Model Diagnosis and Intercomparison (PCMDI) metrics package."

2.) I recommend adding titles to each of the figures.

We have added a title to Fig. 13, and all other figures have titles.

3.) Fig. 7 Caption '"Black, gray, blue, and red curves indicate the satellite-based observations, reanalysis, CMIP5 models, and CMIP6 modes, respectively." I think is intended only for Fig. 6.

Thank you for pointing it out. We have replaced "curves" with "markers" in the caption.

4.) In Table 3, is there not a citation for FracPR?

We have added a reference for FracPRdays in the table.

5.) The U.S. DOE Benchmarking Report is in the reference list, but I do not see it referenced in the text. I think instead you use Pendergrass, 2020?

Thank you for pointing it out. We have cited both references in the revised manuscript.

Overall, I think these results should be published and will be beneficial to the research community, following revision to the text, primarily to address my first item of suggestion.

We are grateful for the comments and suggestions of the reviewers that helped improve the manuscript.

---

## Author Response (AR2)

**Comments to the author:**

Dear author,

Thank you very much for your revised manuscript that, from my point of view, fully answers the referee's remark.

I just have a very minor comment ; I think the « number of the wettest days » on pp.4, 6 and 8 should be replaced by « number of wettest days » (as is used elsewhere in the text). It may seem as a detail but it took me a while to understand Fig 1b and I think I would have been quicker without the « the »!

With best regards,
Sophie

**Response to the editor:**

We appreciate your role as the editor of this manuscript and for providing valuable feedback. Following your suggestion, we have replaced "number of the wettest days" with "number of wettest days" in the text and Fig.1b.

**Notification to the authors:**

The table is included as figure (Figure 13). Please re-label this as table and the references in the manuscript text must be adjusted accordingly. If the colour spectrum of these tables is necessary and cannot be exchanged for footnotes, bold, or italic, then the table must be inserted as an image, but still be called a table.

**Response:**

We propose that Figure 13 be classified as a figure, given that the primary information it conveys is through color shading, while the numbers in boxes serve as annotations. May we keep Figure 13 as a figure?